# Treatment decision making (TDM): a qualitative study exploring the perspectives of patients with chronic haematological cancers

Dorothy McCaughan [1] , Eve Roman [1] , Alexandra Smith [1] , Russell Patmore,[2] Debra Howell [1]

¹Health Sciences, University of York, York, UK
²Queens Centre for Oncology, Castle Hill Hospital, Cottingham, UK

**Correspondence to**
Dr Debra Howell;
debra.howell@york.ac.uk

## ABSTRACT

**Objectives** Haematological malignancies are the fifth most common cancer in the UK, with chronic subtypes comprising around a third of all new diagnoses. These complex diseases have some similarities with other cancers, but often require different management. Surgical resection is not possible, and while some are curable with intensive chemotherapy, most indolent subtypes are managed with non-aggressive intermittent or continuous treatment, often over many years. Little is known about the views of patients with chronic haematological cancers regarding treatment decision making (TDM), a deficit our study aimed to address.

**Setting and design** Set within the Haematological Malignancy Research Network (HMRN: www.hmrn. org), an ongoing population-based cohort that provides infrastructure to support evidence-based research, HMRN data were augmented by qualitative information from in-depth interviews. Data were analysed for thematic content, combining inductive and deductive approaches. Interpretation involved seeking meaning, salience and connections within data.

**Participants** Thirty-five patients with four chronic subtypes: chronic lymphocytic leukaemia, follicular lymphoma, marginal zone lymphoma, and myeloma. Ten relatives were present and contributed to varying extents.

**Results** Five themes were discerned: (1) Preference for clinician recommendations; (2) Factors implicated in patient involvement in TDM; (3) Perceptions of proactive/non-proactive approaches to TDM; (4) Experiences of TDM at various points in the disease trajectory; (5) Support from others. Our principal finding relates to a strong preference among interviewees for treatment recommendations from haematologists, based on trust in their expertise and perceptions of empathetic patient–clinician relationships.

**Conclusion** Interviewees wanted to be involved in TDM to varying extents, contingent on complex, inter-related factors, that are dynamic and subject to change according to differing clinical and personal contexts. Patients may benefit from clinicians assessing their shifting preferences for involvement on multiple occasions. Strong preferences for acceptance of recommendations was associated with cancer complexity, trust in clinician expertise and positive perceptions of patient–clinician relationships.

## Strengths and limitations of this study

► Set within the infrastructure of an established population-based study, this is (to our knowledge) the first UK study to specifically explore treatment decision making involvement in patients with chronic haematological malignancies.

► Interview data supplemented robust clinical information routinely collected as part of the broader cohort.

► The sample size (35 patients) enabled in-depth exploration of the research questions, enhanced by contributions from relatives/carers.

► Purposive sampling ensured inclusion of individuals with differing demographic and disease profiles, diagnosed at different time points on the clinical pathway.

► The views of participants are unlikely to reflect those of the entire patient population, and dedicated studies of people from black and minority ethnic backgrounds and those with low literacy levels are required.

## BACKGROUND

Haematological malignancies are the fifth most common cancer in the UK.[1–3] They are complex diseases, and although they have some similarities with other cancers, they often require different management. Surgery is not an option, and while some subtypes are curable with intensive chemotherapy and periods of hospitalisation, other more chronic subtypes are not. These more indolent malignancies, which account for around 30% of all newly diagnosed haematological cancers,[4 5] can, however, often be controlled for long periods; typically (but not always) following remitting/relapsing pathways wherein periods of chemotherapy are interspersed with active monitoring, known as Watch and Wait (W&W).[6 7] These pathways are associated with variations in the extent of hospital activity, with periods of disease

progression, relapse and treatment requiring more consultations and decisions, and W&W having fewer contact points, often months apart. Haemato-oncology is a rapidly changing field, and the past decade has seen the development of innovative treatment regimens that include targeted therapies, along with new and established chemotherapies, radiotherapy and haematopoietic stem cell transplantation (SCT).[8] While obviously beneficial, these developments increase treatment decisional complexity for clinicians and patients.[9]

Shared decision making (SDM), seen as a hallmark of quality care, is part of a broader concept of patient-centred care which considers individual preferences, needs and values, with the aim of ensuring patient values guide all clinical decisions.[10–12] Steps integral to SDM include the clinician informing the patient of treatment options and the need for a decision; discussion between patient and clinician of each option; and the clinician supporting the patient to consider each option, before reaching an informed decision.[13–15] Elicitation of patient preferences is considered central to effective SDM.[16 17] When multiple treatment options exist, each of which may be associated with different risks, benefits and quality of life implications, adherence to SDM steps can result in a decision that is optimal for the individual patient (and their relatives/carers). If evidence for a specific treatment is strong, the clinician may make a recommendation, informing the patient of their reasoning, which the patient may accept or decline.[15 18] Audiotaped consultations between 236 patients and 40 haematologists in the USA[19] revealed that patient preferences were not commonly elicited, and that treatment recommendations were provided by haematologists in 97% of consultations. More recent studies, (mainly surveys), indicate increasing desire among patients with haematological cancers for involvement in treatment decisions,[20–22] but highlight dissatisfaction with information received. A recent thematic review[15] of 18 haematological studies, indicates three critical, but modifiable, barriers to patient centred communication, a prerequisite for SDM: insufficient information exchange, treatment goal misalignment and discordant (patient preferred and actual) role preferences in decision making. Despite the recognised need for qualitative research to better understand and contextualise preferences for involvement in decision making among haematology patients,[15] few qualitative studies have been conducted.

Our study was designed to investigate the perspectives of patients with one of four haematological cancer subtypes: chronic lymphocytic leukaemia (CLL), follicular lymphoma (FL), marginal zone lymphoma (MZL) and myeloma. Specifically, we aimed to explore patients' understanding and experiences of involvement in treatment decision making (TDM), and to identify factors promoting or impeding this process. This study constitutes one strand of a larger programme of work designed to provide evidence-based information about the management and experiences of the general population of patients with chronic haematological malignancies. A suite of further papers are under preparation, which address information seeking and sharing, patients' experiences of disease management and needs for support.

## METHODS

Methods are reported in accordance with the Consolidated Criteria for Reporting Qualitative Research Checklist.[23]

### Study design

A qualitative, descriptive study,[24 25] utilising semistructured in-depth interviews.

### Sample and setting

The study was conducted within the UK's Haematology Malignancy Research Network (HMRN: www.hmrn.org), a unique collaboration between university academics, National Health Service (NHS) clinicians, and patients and carers that facilitates research using various methods, with the purpose of generating evidence to underpin improvements in clinical practice.[6 7]

HMRN includes a population of around 4 million people in Yorkshire and Humberside and has a similar age and sex profile to the UK and a comparable socioeconomic and urban/rural distribution.

Sampling in qualitative research aims to acquire information that is useful for understanding the complexity, depth, variation or context surrounding a phenomenon.[26 27] We, therefore, aimed to capture a broad range of diverse experiences; initial criteria included proximity to the median diagnostic age for each disease subtype, with variation by gender, ethnicity and postcode, as well as time since diagnosis. Over time, our sampling strategy evolved to select individuals across broader demographic categories (eg, those relatively young when first diagnosed), and to ensure inclusion of participants at different time points on the clinical pathway. Patients were asked to invite a relative to join the interview, if they wished, and ten contributed to varying extents. Details of the study sample can be seen in table 1.

### Data collection

Using existing links with NHS teams, initial checks ensured patients were alive and well enough to participate. Potential subjects were sent information about the study, and a letter inviting them to be interviewed, with a reply slip and pre-paid envelope. The researcher's contact details and free-phone number were included so that patients could discuss the study before deciding whether to take part. Thirty-five interviews were conducted (DM) between February and October, 2019; and 10 relatives participated. The majority took place privately, in patients' homes and lasted around 40–90 min. Assurances of confidentiality and anonymity were given to participants and written consent was obtained from patients and relatives. During interview, a (small) number of patients became

**Table 1** Characteristics of interviewees

| ID | Diagnosis | Age-range at interview (years) | Years since diagnosis | Known treatment line(s) at interview (via Haematolgoical Malignancy Research Network data)*† | | | | | |
|---|---|---|---|---|---|---|---|---|---|
| | | | | 1 | 2 | 3 | 4 | 5 | 6 |
| P1 | CLL | 60–70 | 4 | Observation | – | – | – | – | – |
| P2 | MZL | 60–70 | 15 | Observation | Chemotx | Observation | – | – | – |
| P3 | CLL | 60–70 | 22 | Observation | Chemotx | Observation | – | – | – |
| P4 | MZL | 60–70 | 2 | Observation | Chemotx | – | – | – | – |
| P5 | MZL | 50–60 | 8 | HPE | Observation | – | – | – | – |
| P6‡ | CLL | 70–80 | 8 | Observation | Chemotx | Observation | – | – | – |
| P7‡ | CLL | 60–70 | 6 | Observation | Chemotx | Observation | – | – | – |
| P8 | FL | 70–80 | 3 | Chemotx | Radiotx | Observation | – | – | – |
| P9 | CLL | 80–90 | 5 | Observation | Chemotx | – | – | – | – |
| P10 | FL | 70–80 | 8 | Observation | Chemotx | Chemotx | Chemotx | – | – |
| P11 | Myeloma | 60–70 | 7 | Observation | Chemotx | Observation | – | – | – |
| P12 | MZL | 70–80 | 5 | Observation | Chemotx | – | – | – | – |
| P13 | CLL | 50–60 | 1 | Observation | – | – | – | – | – |
| P14 | Myeloma | 60–70 | 4 | Steroids | Radiotx | Chemotx | Chemotx | Chemotx | SCT |
| P15 | FL | 70–80 | 3 | Observation | Chemotx | – | – | – | – |
| P16 | Myeloma | 60–70 | 2 | Chemotx | Chemotx | Chemotx | SCT | Observation | – |
| P17‡ | FL | 60–70 | 3 | Observation | Clinical trial | Observation | – | – | – |
| P18 | Myeloma | 60–70 | 3 | Chemotx | Chemotx | Chemotx | SCT | Observation | – |
| P19 | FL | 50–60 | 3 | Steroids | Chemotx | Chemotx | Observation | – | – |
| P20‡ | CLL | 70–80 | 4 | Observation | – | – | – | – | – |
| P21‡ | Myeloma | 70–80 | 3 | Steroids | Chemotx | Chemotx | Chemotx | SCT | – |
| P22‡ | CLL | 70–80 | 3 | Observation | Clinical trial | Observation | – | – | – |
| P23 | Myeloma | 60–70 | 3 | Observation | Observation | – | – | – | – |
| P24 | FL | 50–60 | 4 | Steroids | Chemotx | Radiotx | Observation | – | – |
| P25 | FL | 60–70 | 4 | Chemotx | Chemotx | – | – | – | – |
| P26 | Myeloma | 70–80 | 4 | Observation | – | – | – | – | – |
| P27‡ | CLL | 70–80 | 4 | Chemotx | Observation | – | – | – | – |
| P28 | Myeloma | 60–70 | 4 | Steroids | Chemotx | Chemotx | SCT | Clinical trial | Chemotx |
| P29 | CLL | 70–80 | 3 | Clinical trial | Observation | – | – | – | – |
| P30‡ | Myeloma | 70–80 | 2 | Observation | – | – | – | – | – |
| P31‡ | Myeloma | 70–80 | 2 | Radiotx | Steroids | Chemotx | Observation | – | – |
| P32‡ | MZL | 60–70 | 2 | Observation | Chemotx | Observation | – | – | – |
| P33 | Myeloma | 50–60 | 3 | Chemotx | Chemotx | SCH | Observation | – | – |
| P34 | FL | 50–60 | 4 | Steroids | Chemotx | Chemotx | Chemotx | – | – |

Continued

**Table 1** Continued

| ID | Diagnosis | Age-range at interview (years) | Years since diagnosis | Known treatment line(s) at interview (via Haematolgoical Malignancy Research Network data)*† | | | | | |
|---|---|---|---|---|---|---|---|---|---|
| | | | | 1 | 2 | 3 | 4 | 5 | 6 |
| P35 | Myeloma | 50–60 | 2 | Chemotx | Chemotx | Chemotx | Chemotx | SCT | Observation |

*SCT (all autografts); SCH (shown as SCT did not take place).
†Does not include supportive care (eg, blood product transfusions, plasma exchange, bisphosphonates, cell mobilisation products).
‡Relative present at interview.
Chemotx, chemotherapy; CLL, chronic lymphocytic leukaemia; FL, follicular lymphoma; HPE, *Heliobacter pylori* eradication; MZL, marginal zone lymphoma; Radiotx, radiotherapy; SCH, Stem cell harvest; SCT, stem cell transplant (all autografts).

upset while reflecting on their cancer and its progression. Although these patients were asked if they wanted to pause or discontinue the interview, all wished to continue, some commenting afterwards that the discussion had helped clarify their thinking about their disease. Interviews were digitally recorded and transcribed verbatim, checked for accuracy and anonymised. Recordings and transcriptions were stored in accordance with legally required data protection standards and ethically approved practices. Interviewing continued until it appeared no new information was forthcoming, a signal that data saturation was likely achieved,[28] and the recruitment end-point occurred when preliminary analysis indicated patterns and themes with sufficient data.[29]

Interviews were directed by a semistructured topic guide (online supplemental appendix 1) based on research literature and input from clinicians (haematology specialist consultants and nurses) and piloted with 2 patients (from a haematology cancer support group) to check comprehensiveness and comprehensibility. The guide was modified over time to include new lines of inquiry, and was used flexibly to allow patients to 'tell their story' from diagnosis onwards.

### Data analysis

The analytical approach adopted was qualitative description,[24] based on thematic content analysis.[30] Qualitative description research seeks to discover and understand a phenomenon, a process or the perspectives and worldviews of the people involved.[31–33] Analysis was undertaken by two members of the research team (DM and DH), both experienced in qualitative methods in applied health services research and haematology. Interviews were summarised through dynamic engagement with the dataset, while staying close to participants' accounts.[24] Our aim was to translate the 'raw' data into a coherent depiction of the phenomena under scrutiny.[34] Guided by the research questions, our analysis balanced both inductive and deductive orientations.[35] We familiarised ourselves with the content of the transcripts to identify initial codes (units of meaning) and themes. These were modified and expanded during an interactive and reflexive process of 'interrogating' the data, in the search for common patterns and 'deviant' or 'negative' cases not supporting, or appearing to contradict, patterns or explanations emerging from data analysis.[36] Data were then summarised and compared, within and between cases. Our analysis therefore facilitated data synthesis and interpretation, enabling a detailed and nuanced account of the findings.[34] Analytical rigour was promoted through reflective notes and memos, and discussion of disagreements helped refine the analysis.[37]

### Patient and public Involvement

Patient and public involvement is integral to HMRN and lay-individuals are routinely involved in all research activities. For this particular study, patients and relatives were involved in prioritising aims, preparing the funding

application, attending programme steering committee meetings and the dissemination of findings.

## RESULTS

Thirty-five patients (19 male, 16 female) were interviewed, 10 accompanied by relatives (spouse/partner or other family member). Most were aged 50–70 years; 32 lived with a spouse/partner or other family member and three lived alone. Ten patients had CLL, 8 FL, 12 myeloma and 5 MZL. Prior to interview, patients had experienced different treatment pathways, according to diagnosis and disease progression; seven had started and remained on W&W, while the remainder had been treated at least once, including six who had experienced multiple lines of chemotherapy before progressing to SCT. Patient characteristics and individuals' treatment pathways, ascertained from HMRN routine data collection, and patient self-report, can be found in table 1.

The following five themes were identified: (1) Preference for clinician recommendations; (2) Factors implicated in patient involvement in TDM; (3) Perceptions of proactive/non-proactive approaches to TDM; (4) Experiences of TDM at various points in the disease trajectory; (5) Support from others.

### Theme 1: preference for clinician recommendations

Most of the patients interviewed in our study said they wished to be informed about, and given the opportunity to discuss, treatment benefits, risks and outcomes with their doctor, while expressing a preference for the clinician to make a recommendation. Some said they wanted their consultant to explain the rationale for recommendations, while others said they weren't interested, that they just wanted *'to get it* [treatment] *over and done with'*.

'they spoke to me so much, I feel involved**…** I would hate for them to say, you're going to do that, with no explanation…and that doesn't happen. When I have treatment, I am told why.' (P9)

'I wanted their expertise and their guidance…I felt very involved but I didn't necessarily feel I should be making the ultimate decisions… I wanted them to make the decisions…' (P19)

I would just do what they recommend really' (P23)

'I just like the person telling me what I've got and what they're going to do' (P27)

Preferences for clinician recommendations were said to be based on patients' respect for haematologists' clinical knowledge and expertise, trust in their professional judgement, and faith that they would have the patient's best interests at heart.

'I do have a respect for medical people because they've done all the training, so I would hope they would say… well we think this is probably the best way forward, because I think, well, how would I know?' (P26)

'I've got total faith in what they are doing…I am certainly not capable of making a medical decision on my behalf' (P21)

'the team are making that decision for me, and they're working for me with my best interests at heart' (P13)

Trust in individual haematologists was strongly linked to patient perceptions of (mainly) excellent patient/doctor relationships, characterised by consultants' willingness and ability to: demonstrate empathy (by helping patients feel at ease, and valued as individuals, *'not just a number'*); tailor information to match individual requirements; listen and respond to questions and concerns; initiate and engage patients in open and frank discussion; and impart hope and some positivity when things are not progressing well.

'she [haematologist] explained everything…I could understand everything…there was no stress involved…she seems to be able to ask the right questions and she takes it all in…she tends not to write down until we've finished talking…we just have a chat basically…she is so warm and pleasant, smiling, we have a laugh…I mean you come away feeling elated rather than 'phew' (P10)

'maybe they just think well, if somebody is not asking at the moment, we won't say, because this is too much information. Maybe they take their lead from where you're at…' (P27)

'giving time for people to actually speak…especially when people are ill, to think about things before they ask questions…so listening is probably at the top of the list' (P29)

'let's have everything as open as possible so that everybody who's in the equation knows what's going on or what possibilities are out there for treatments' (P14)

'I was a bit frightened but she said, if you do exactly as we've planned…you will come out of it…I'm so certain of that. Now, when any consultant tells you that, it sort of lifts you' (P9)

Trust was said to be enhanced when patients knew that treatment discussions had included members of the wider haematology team, as happened within the context of multidisciplinary team meetings, if their consultant had sought a second opinion from haematologists specialising in the patient's condition, and when the patient was aware that they were receiving treatment in a recognised centre of excellence.

'these people [haematologists] are much more expert than I'll ever be…the consultant that is making the decision, it's his team. You've been talked about, so it isn't one person making that decision…in the background there's quite a lot of people that are involved and I find that really, really soothing' (P13)

'it's reassuring that it is not just one person making that decision…when he [haematologist] decided I needed to start chemo, he did say he had actually spoke to a colleague of his at [place] University…I think he was a professor…and told him my symptoms, and the professor had also said, well, yes, I think it's time to start treatment…' (P4)

I knew they were a centre of excellence and they had very high standards…so, I just felt very…secure' (P19)

Interestingly, some people revealed awareness of 'their' consultant's clinical and academic credentials.

'he [patient's consultant] was the lead consultant for the team so I thought, well that is a good recommendation' (P24)

'and Prof [name] who I've seen perhaps 4 times and everyone wants to see Prof [name], because he's got the biggest brain [laughs] (P28)

Only one study participant (P3, with CLL) mentioned having themselves sought a second opinion, commenting that far from taking offence, his consultant assisted him in identifying appropriate specialists to contact.

'he [haematologist] didn't take that as a personal insult…you feel more confident…going to see one of the top specialists [second opinion]…you're going to see a range of experts who have agreed this is the best option…you are being backed by a range of experts' (P3)

### Theme 2: factors implicated in patient involvement in TDM
The extent to which participants in our study said they were involved in TDM appeared to vary according to a range of inter-related individual ('*everybody is different*') and contextual factors, including decision complexity; individuals' ability and desire to access, interpret, and retain information about their cancer; personal preferences and values; patients' physical and/or emotional state, and coping mechanisms; and the level of support from others (elaborated in Theme 5). These factors were drawn together to develop this theme and are summarised in figure 1, with quotes (below) illustrating each component. While some patients wanted to hand over much, or all, responsibility to clinicians, others took steps to enable participation in discussions; many interviewees clustered somewhere between, reporting varying levels of engagement/non-involvement, dependent on circumstances.

(Individual differences):

'every patient is different…there probably are patients who just rely on their specialist…most are happy to just 'get on with it gov', whereas I was always asking questions and that's what I would recommend to other patients, is try to understand and be

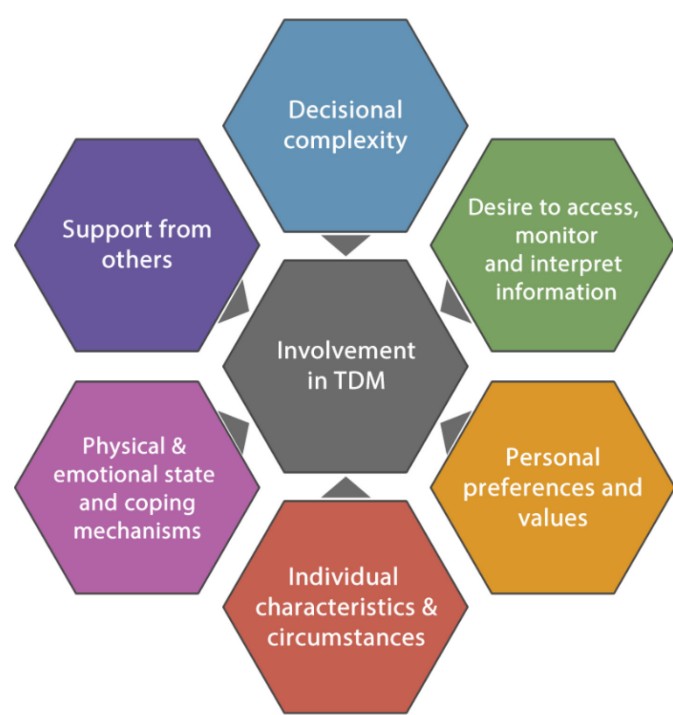

**Figure 1** Factors implicated in patient involvement in treatment decisions. TDM, treatment decision making.

proactive… that way at least when [there's] any option to decide if I can feel confident that's the best available for me' (P3)

(Decisional complexity):

'in an ideal world it would be myself, me and the doctors [who share TDM] because to me, they've got so much knowledge about this thing that I don't know anything about… I'd just like to have something [simple] like, Oh, you've got appendicitis'…but I'm not in that situation. It is complex…they don't know all the answers…because there's different types of it [myeloma] and I don't fully get the reasons why everyone reacts differently (to treatment)' (P35)

(Information access):

'I'm not an internet person…it annoys me, the internet' (P15)

(Information understanding):

'Not everyone is equipped to sort of read some of the literature because…it is a little bit challenging…and time consuming' (R, P6)

(Information retention):

'We were given some leaflets…but I'm a bit fuzzy… since I've had my cancer my memory has gone to shot' (P16)

(Physical/emotional state):

'I was so rock bottom, I guess I just went along with it all…they knew what they were doing' (P8)

(Coping mechanism):

'there are some people who want to know every detail about every treatment and how it affects them…I'm scared to do that' (P35)

(Personal preference (for proactive role)):

'PatientView [portal for accessing patient electronic record]…that's fantastic because I can see things, so your platelets, white [blood cells]…haemoglobin… it's a fantastic bit of information for me prior to my consultant appointment…having that extra information, for me, is very valuable…I can point out…to the consultant…so what's this about…you know, why's that gone up?' (P28)

(Personal circumstances):

'your mind is concentrating on other things…is it life threatening…am I going to be able to work anymore…have I got to retire…you've got the financial aspects…making sure your family is looked after… your brain is working on so many different levels… you tend to accept what's [treatment] being given' (P16)

(Level of support):

'that's what I miss, because I have to do all my own research…and sometimes I think I just wish I had that person… [for support]' (P33)

### Theme 3: perceptions of proactive/non-proactive approaches to TDM

During interview, a number of patients articulated views relating to proactive involvement in TDM, within the context of their stated preference for acceptance of haematologists' recommendations, while others described factors likely to impede or diminish active engagement in TDM. Factors associated with proactive and non-proactive approaches to decisions are summarised in figure 2.

Those who chose to be proactive described the need for certain resources (internet, time), skills (retrieving/ interpreting relevant information) and a high level of personal commitment. This group also tended to access information from many sources, including UK charities (eg, Myeloma UK and CLL Support Association), national and local support groups, on-line patient forums (mainly US based), clinical nurse specialists and peer-reviewed articles. Some patients described how internet access was essential, along with the ability to distinguish between *'authoritative'* and *'dreadful'* websites. Taking time to read about blood/lymphatic systems was seen as a necessary pre-requisite for understanding specific diseases. Patients who sought in-depth information from journal publications tended to have some prior knowledge which facilitated their understanding, up to the point when they encountered 'the buffers' of what they could comprehend. Even these patients, who were well equipped to retrieve and assimilate information, reached a limit to their capacity to understand information relating to their haematological cancer at a specific juncture; other study participants indicated that they had difficulty understanding much/most of the information they encountered.

'I did intelligent searches…I was confident and used to reading science papers…' (R, P6)

'I had read on the internet all about these different prognostic indexes and stuff…when I was told prognosis was 5 years…I already knew what prognosis meant' (P34)

**Proactive approach**

Strong motivation to be involved in TDM

Requests, retrieves, reads and interprets information from various sources

Gathers and reflects on information before, during and after consultations

Formulates clear questions, writes these down to ask in clinic, and records responses

Monitors and interprets blood results

Accesses and reads letters to GP

Often supported by other(s) who may act as 'information broker' and advisor/ counsellor

Has confidence to 'speak up' to ask for further explanation and discussion, or a second opinion

Asks for time to reflect on proposed treatment

**Non-proactive approach**

Preference for little/no involvement in TDM

Reliance on haematology clinicians for information

Information perceived as difficult to comprehend

Unsure what questions to ask clinicians

No desire to monitor or interpret blood results

Access to information (e.g. GP letters) not pursued

Lacks support from others

Disinclined to seek information about treatment outcomes, disease progression, prognosis and survival

**Figure 2** Continuum of characteristics associated with proactive and non-proactive approaches to involvement in treatment decisions. GP, general practitioner; TDM, treatment decision making.

'I'd be looking at Lancet type papers…PubMed' (P24)

'it turned out the free light chains had rocketed… so that ruled out second stem cell transplant' (P18)

'it's good to know statistics…I wanted to know about survival figures…I realised that I was looking at the population as a whole, but then I thought I'd look at younger people…' (P3)

'it's challenging because you will reach the buffers at some point, when you think, that's just a little too difficult to understand' (R, P6)

Proactive patients described preparing for consultations. Despite busy clinics, patients said doctors rarely made them feel time was constrained, usually invited questions, and took time to respond. Some patients prepared questions beforehand, a few noted the consultant's response(s), with some having relatives acting as 'scribe'. Keeping a record or graph of blood test results, used as a trigger for questions, and maintaining a diary of disease progress and treatments were common. These patients sought results from investigations (blood tests, bone marrow biopsies, X-rays, CT scans, etc) and often asked consultants to explain their significance. They also read about the risks, benefits and possible side effects of treatments, to equip themselves to engage in further discussion. Being prepared to 'speak up' during consultations was said to be important, though was recognised to be easier for some than others: *'you have to be reasonably assertive, which is maybe not my strength, but I force myself in those situations where I know how important it is'* (P3). Only one patient (P6) and their spouse (R,P6) reported 'speaking up' to express a preference for a treatment other than the one offered by their consultant, based on their own *'research'*, that suggested it would be less detrimental to the patient's quality of life: *'we knew the gold standard was treatment with Fludarabine, FCR as they call it, it really wasn't recommended for people aged over 65, but nonetheless it was on offer to us…we actually then chose the Bendamustine route…the gentle one…which was a good alternative…we'd done our research, we made an intelligent decision'* (R, P6).

Many respondents expressed little or no desire to engage in active TDM, preferring to rely on clinicians' expertise, as reflected, for example, in the following comments: *'I don't want to go into details of things…the people at the hospital are there to help me…I put my absolute trust in them'* (P10); *'I went along with everything that I was asked to do because I have complete confidence in how they were handling the situation for me'* (P14).

Factors that could impede patients adopting a proactive approach included difficulties accessing and/or fully understanding information, a disinclination to dwell on disease progression and prognosis (*'in denial'*: P35), or patients feeling unwell, anxious, overwhelmed, or unsupported.

'I've got a little tablet but I very rarely use it. I don't like to, I'm not computer literate' (P30)

'the specialist at the hospital said, it's only forty-four, which meant nothing to me, forty-four what? She didn't say what and I never asked her.' (P31)

'sometimes they come out with all these big words and then you think, I'm not sure what that word is' (P24)

'I have glanced through it [booklet about myeloma] but it's a bit too high a level…it needs to be basic' (P21)

'I saw this research about myeloma…and when I read the title I wasn't sure whether it was going to be that helpful, because I didn't understand what it was really…' (P35)

'psychologically, I brush it under the carpet a bit because I know it [chemotherapy] is not imminent' (P5)

'I've just gone back to being anxious again…it's just horrible being in this position where you know it's [paraprotein level] is creeping up' (P35)

'I am strong, you know, but it just becomes too much… I want somebody just to hold my hand and go, I'm going to sort that for you.' (P33)

### Theme 4: experiences of TDM at various points in the disease trajectory

Patients' accounts indicated that their ability and desire to be involved in TDM could vary or indeed be curtailed at different points in the disease trajectory. Most participants described feeling deeply shocked, upset and anxious when first informed of their diagnosis (*'like a huge bombshell'*), to the extent that they could not absorb what was being said to them or think of questions to ask (*'it's as if your brain switched off'*), reactions likely to compromise ability to participate in discussions about treatment options.

'we were floored…[by diagnosis]…it hit us out of the blue…you can't think of questions in that short space of time…we were speechless…we just sat there in shock…you hear the word cancer…' (P32)

'it was a shock when he mentioned the word cancer… it just sort of shut me down…he was willing to give me information but at that time, I just couldn't process it' (P4)

Some patients on W&W, who subsequently went on to require treatment, reported little or no involvement in TDM, as there was only a single relevant option. In this context the decision was said to be 'automatic' and that it would be the 'standard' treatment. Some of these participants said they would have welcomed having more time to discuss the proposed treatment with family members, and to consider whether or not to accept it; a few recalled their consultant strongly recommending that they accept the treatment offered.

'It was a fait accompli really' (P15)

'I was told that I had to have chemotherapy, 6 chemo-therapies…one every 3 weeks and radiotherapy after that' (P8)

'I was very worried and scared and thought I might choose not to have the treatment…because I felt physically very well…why subject myself to being ill' (P1)

'[consultant said] "I must tell you that even though you don't want to go on chemo….my recommenda-tion is that you do"' (P9).

Opportunities for involvement in decision making appeared circumscribed when urgent treatment was required. Examples came from P19, who was admitted to hospital acutely ill and subsequently diagnosed with FL, from P24, who said he was diagnosed with *'a very unusual lymphoma',* and from P21, with myeloma. In these instances, each patient referred to TDM occurring at speed, largely without their involvement, as they relied on the expertise of the clinical team to make the 'best' decisions on their behalf.

'There weren't really any options…it was a case of this is what is best for you… I didn't have time to think about it… it [the cancer] was so advanced…they de-cided… I don't think I was really involved in that…I wanted them to guide me and make the decisions' (P19)

'it was a very unusual type of lymphoma…there was a huge team involved in it all and even throughout their decision making they changed from what were originally going to do, which was radiotherapy…they were just going to blitz it, but my kidney would have been in the way, so they then decided to go down the avenue of chemotherapy and the monoclonal…' (P24)

'they [doctors] decided on stem cell transplant straightaway' (P21)

Disease progression resulted in some patients with myeloma feeling overwhelmed when faced with diffi-cult treatment decisions, and unable to choose between options. Factors compounding this included the intensive nature of proposed treatments (such as SCT) and their impact on quality of life; the limited 'returns' that some treatments seemed to offer, compared with the conse-quences of associated risks, such as infection; and the uncertainty and unpredictability of outcomes.

'Honestly, my head is exploding with all this…it's just like a big crushing thing to me…I think I am quite strong but this is doing me in' (P35)

'the big one [decision] was the second stem cell trans-plant…I was really struggling to make the decision as to whether I wanted to go for it' (P18)

'they [doctors] sort of said, well the average remission after the stem cell transplant is, I think either 12–24 months, or 18–24 months, something like that, and

there was I thinking, right I'm going for the 10-year option, so that was quite a shock' (P18)

'Myeloma is a very individual disease. You get the same treatment, same, same this, same that but you have different outcomes and things' (P28)

## Theme 5: support from others

Our interviewees said that their relatives often accom-panied them to clinical consultations, and they were portrayed as playing an important, and sometimes crucial, role when treatment options were being discussed and considered: *'she [patient's wife] guided me…she would trans-late…she would talk to the ward people…'* (P7). Patients benefited from *'going over'* information from their haema-tologist with their relative following consultations, and discussing the details of any proposed treatment, side effects, and implications for quality of life with them: *'we'd come home and discuss it [information from the consultant] what she said about so and so…having two sets of ears helped'* (P11). Some patients (eg, those who mentioned difficulties processing and retaining information) described relying heavily on support from relatives (spouse/partner/adult child/sibling) for all interactions with clinicians: *'my daughter always takes notes…so when we come away we can go through them…they are quite happy with that…if I go on my own, I would retain some of it and I'd probably forget some of it… my daughter knows all about my treatment'* (P9).

Relatives' roles encompassed gathering and inter-preting information, acting as a sounding board and providing practical and emotional support to patients preparing for, and undergoing, treatment.

'we were proactive and looked for information' (R,P7)

'we'd [patient and partner] talked through it [deci-sion related to stem cell transplant] over a number of weeks really…we'd come back to it…there's this factor and that factor…' (P18)

'there are times when my wife has come to the res-cue…particularly if I get an infection…I'm thankful there is somebody else around' (P11)

Many patients and relatives used the pronoun *'we'* throughout the interview, though there was general agreement that the final decision of whether or not to accept a treatment, was/would be the patients.

'we'd both be involved…it wouldn't be one partner on their own, it would always be the two of us involved together' (P21)

'he [husband] doesn't sway me, he leaves it very much up to me, he wouldn't persuade me…' (R, P1)

'I'm as involved as much as I can be but at the end of the day it's [patient's name]… he has to make the decision, I can't make it for him' (R, P31)

The importance of emotional support provided by a spouse/partner to patients experiencing anxiety and inner turmoil associated with diagnosis and TDM was

repeatedly emphasised: '*she's [patient's wife] been with me at every step*' (P27); '*you need somebody to be the rock…to take the strain off you*' (P16); '*I don't know how I could have coped without him [husband]*' (P6).

Three study participants living alone commented that they often felt unsupported in relation to their cancer, feelings that were heightened when treatment decisions arose: '*I knew chemo was what I'd have to have but when he* [consultant] *told me, that shocked me, and I was so upset and scared…really scared*' (P4). In the absence of a spouse/partner they turned to friends or close family members for advice and support, but did not want to be a burden to them: '*I've got a very good friend…we've known each other 50 years…I would never have got through it without her…you need support…somebody talk to…but she has her own family…I don't like to be a burden to anybody*' (P8). Two of the three participants without a spouse/partner mentioned seeking formal psychological support, so that they might have the opportunity to share and discuss their feelings and experiences.

Eight of the 35 participants had joined a formal support group. Reported benefits included hearing about up-to-date research findings from invited speakers (often clinicians), and having the opportunity to talk to other people about their experiences, which helped some patients think through advantages and disadvantages of their own options for treatment. Those who preferred not to join a group ('*not for me*'), ('*I'm a very private person*'), often sought one-to-one support, through meetings with former patients arranged by clinicians, personal contacts with someone with a similar diagnosis, and/or via on-line patient forums.

## DISCUSSION

This qualitative study provides new insights into patient perspectives of involvement in treatment decisions, their views on proactive engagement, and the role of others in supporting decision making. The findings reflect the broad array of interconnected mechanisms at play in shared decision making,[38] and its complex and dynamic nature.[39–41] The principal finding among our interviewees was the strong preference for treatment recommendations to be provided by haematologists, based on trust in their clinical expertise and perceptions of empathetic patient/clinician relationships. Most participants expected/wanted an explanation from their clinician about the rationale for treatment decisions, including details of possible risks and benefits, but did not wish and/or felt ill equipped, to make decisions on their own behalf. This finding does not align with some recent reports of the growing desire for involvement among patients with haematological malignancies,[20–22] though seems unsurprising within the detailed contexts depicted by our interviewees. Participants with myeloma, who had most decisions to make due to the nature of their cancer, tended to experience decision making as challenging, struggling to weigh up the risks, adverse impacts on quality of life, and prognostic uncertainty associated with different treatments

options; which left them inclined to follow clinician recommendations. Patients with FL also reported trusting clinicians to make decisions on their behalf, particularly when they were acutely ill at diagnosis (as may occur in some of these typically indolent cancers), and unable to participate in discussions. Likewise, participants with CLL were mainly inclined to entrust decisions to clinicians; however, some expressed dissatisfaction that they had not been given the opportunity to fully consider, and accept, or decline, treatment. Dissatisfaction among this group of patients was possibly, in part, attributable to the fact that patients with CLL often had infrequent contact with haematology HCPs during long periods of observation, with months sometimes elapsing between appointments.

Drawing on results from their systematic review of the literature relating to physician views of SDM, Pollard *et al*[42] comment that physicians tend to express support for SDM in situations where they do not feel strongly about one treatment alternative, but are less supportive of SDM in situations where compelling, or well evidenced, clinical practice guidelines exist in favour of one treatment over another. In such instances, the decision is not one of selecting between options, but rather whether the patient chooses to accept or decline treatment. This review includes results from an interview study with 20 physicians working in five different settings,[43] that found support for SDM was most common among those who had received communication skills training in this area; and Rocque *et al*[44] suggest that multilevel education programmes, targeting patients with CLL and their clinicians, may improve patient participation in decision making.

Patient involvement in decision making has been linked to improved care experiences and better health outcomes, yet the desire for this has been shown to vary by individuals, number of treatment options and treatment certainty.[45] That patients with haematological malignancies, which are characterised by uncertain trajectories, indistinct transitions and prognostic uncertainty[7 46 47] and novel and evolving treatments, may prefer to defer decisions to specialists, whom they trust, is understandable. Our findings resonate with qualitative studies in Germany[48] and Denmark,[49] that combine interviews with extensive observation of consultations with clinicians and patients with cancer, and which show that most of the time physicians made treatment decisions alone, or with colleagues, with little patient involvement. None of our participants recalled clinicians formally eliciting their preferences regarding decision making, as is recommended,[50 51] yet most felt as involved in this process as they wanted to be, through discussions of treatment options, and clinicians taking time to talk and listen to them, address their concerns, answer questions and offer explanations. Many of the patients in our study clearly felt strong bonds with their specialists, as noted elsewhere,[47] arising from sustained contact in clinic or during hospital admissions. A meta-ethnography of quantitative and qualitative studies[52] underscores the importance patients place on being in a caring relationship with clinicians, which may preclude the need to seek detailed information.

Among our participants was a small number (n=5) who shared certain characteristics (male, (mainly) younger, educated to degree level and highly motivated), and who had adopted a proactive approach to TDM. These individuals generally felt confident interpreting complex information, and were prepared to 'speak up' to obtain further explanation from clinicians; nonetheless, limits to understanding were perceived as constraining their ability to make fully informed decisions. Loh et al[53] caution against predicting preference for decisional involvement of patients with haematological cancers, based on age or characteristics such as educational attainment, suggesting instead that this should be assessed periodically, as part of decision making encounters.

While many respondents preferred little or no involvement in TDM, there were some whose engagement was hindered by the provision of information that did not match their needs, leaving them feeling ill equipped to deal with the complex nature of this material. Clinician assessment of individual health literacy (capacity to access, process and interpret information) is therefore an essential component of SDM, as is noted by others.[15] Providing patients with information that is comprehensible, tailored to their needs and which does not overwhelm,[54] can be challenging. Strategies for 'drip-feeding'[54] information, are to be preferred to a one-way-flow of information from clinician to patient ('broadcasting'), which is regarded as suboptimal.[55] Interestingly, insufficient time for information sharing and discussion during consultations was not generally perceived as a problem in our study, though is reported elsewhere.[15 20 48]

During interview, patients often recalled feelings of profound shock on hearing their diagnosis, reactions that could act as a barrier to meaningful discussion about treatment. A qualitative study[56] including 32 patients with acute myeloid leukaemia revealed similar findings; highlighting the importance of clinicians managing the amount of information patients are ready to receive,[54] both at diagnosis and other timepoints; for example, initiation of treatment after W&W.

The important role played by relatives and others in supporting patients during consultations when treatment options are discussed was very apparent in our study, reflecting results from a systematic review[57] of patient–physician–companion communication that shows companions as instrumental in information transfer and provision of emotional support. Unaccompanied patients in our study said they particularly benefitted from a nurse taking notes of what was said during consultation, and 'talking through' the record with them afterwards. Interviewees without a spouse/partner expressed concerns about burdening friends and family members with their needs for practical and emotional support. While some patients may be reluctant to broach the topic of their cancer with others, or find it difficult to talk about it, they are likely to benefit from discussing treatment options/decisions with someone close to them, who is familiar with their personal preferences and circumstances; haematology doctors and nurses could take time to help patients identify such individuals, and encourage patients to draw on their support.

Coping with disease progression and prognostic uncertainty was said to be particularly difficult by our participants with myeloma. Treatment with SCT (or the newer CAR T-cell therapy) can affect patients physically, psychologically and financially,[58–60] and formal psychological support may be beneficial.[61 62] Several of those considering SCT valued one-to-one support, as also noted by Tariman.[20] Furthermore, it has been suggested that information about 'personal experiences' can complement 'general facts', and contribute to decision support in various ways; for example, by helping people clarify their own values and reasoning, either by suggesting different ways of thinking and/or by providing a 'sounding board' against which to test their own ideas.[63]

## Strengths and limitations

As far as we are aware, this is the first UK study to specifically explore involvement in TDM in patients with haematological malignancies. Our sample size of 35 enabled in-depth exploration of the research questions, and use of semistructured interviews allowed participants to focus on issues they themselves considered significant. Our sampling framework ensured 'key informants' were interviewed from targeted disease subtypes, both sexes and various age groups. As the diseases included are typically relapsing-remitting conditions, we felt it important to include patients whose perceptions may have altered over time, during prolonged W&W or following treatment, to capture as broad a range of views as possible. To counteract the influence of memory, we also invited some recently diagnosed patients to take part; reference to patient diaries and contributions from relatives also enhanced recall. Relatives' participation enhanced the quality of the data collected, through contribution of their own perspectives, prompting patients, and, on occasion, corroboration and/or clarification of patients' accounts.

Attempts to recruit patients from minority ethnic backgrounds were unfortunately unsuccessful. As Morse[64] has highlighted how merging data from a small number of such participants can result in loss of cultural differences when analysed alongside the remainder, who share a single identity, we recommend future in-depth studies, dedicated to those whose heritage differs from participants in our own study. Furthermore, fewer people living in more deprived areas agreed to take part, compared with those in affluent areas. Consequently, a further limitation was our inability to recruit patients with low levels of literacy, who may have been deterred because the study invitation and information was provided in writing. We, therefore, recognise that the views of our participants are unlikely to reflect those of the entire population with the diseases of interest. Nonetheless, it is highly likely that a large proportion of our findings are transferable to other UK areas, and also countries with similar healthcare infrastructure and universal healthcare coverage.

## Clinical Implications

Our findings suggest that our interviewees varied in their preference for involvement in TDM according to intrinsic, contextual and disease-related factors, requiring clinicians to assess individuals' preferences for engagement at multiple time points over the course of their haematological cancer pathway. Fisher *et al*[11] comment that clinicians who clarify patients' preferences and ensure they are informed about their options, are sharing the deliberation aspect of decision making, even if the doctor ultimately provides a strong recommendation. Entrusting clinical staff to make recommendations does not appear to diminish patients' desire for discussion of possible options, and for provision of relevant information that matches their individual needs. Empathetic relationships with clinicians seem highly valued by patients, and appear conducive to engagement in TDM. Deliberation of treatment options can be highly distressing for some patients, and those lacking support from family members/others may benefit from formal assessment and referral for psychological support.

## CONCLUSION

This study revealed that patients with haematological cancers may wish to be involved in TDM to varying extents, contingent on complex, inter-related factors, that are dynamic and subject to change according to clinical and personal contexts. Overall, our interviewees expressed a strong preference for acceptance of clinician recommendations, linked to disease complexity, patients' trust in clinician expertise, and perceptions of trusted patient–clinician relationships.

**Acknowledgements** We wish to thank the study participants who took part in an interview and shared sensitive and emotive issues.

**Contributors** DH, ER, AS and RP designed the study. AS identified potential participants and mapped pathways. DH and DM recruited the study participants, and DM conducted interviews. Transcripts were coded and analysed by DM with discussion/input from DH. DM wrote the first draft of the manuscript. DH, ER and AS revised the manuscript. RP commented on the clinical aspects of the study. All authors read and approved the final version. DH is guarantor for the article.

**Funding** This work was supported by the NIHR via a PGfAR: RP-PG-0613-2002, Cancer Research UK: 29685, and Blood Cancer UK (formerly Bloodwise): 15037. None of the funding bodies were involved in the design of the study, nor in the collection, analysis, interpretation and reporting of data; the views expressed here are those of the authors and do not necessarily reflect those of the funder.

**Competing interests** None declared.

**Patient and public involvement** Patients and/or the public were involved in the design, or conduct, or reporting, or dissemination plans of this research. Refer to the Methods section for further details.

**Patient consent for publication** Not applicable.

**Ethics approval** This study involves human participants and was approved by Ethics Committee providing approval: London, City and East: REC:16/ LO/0740. Participants gave informed consent to participate in the study before taking part.

**Provenance and peer review** Not commissioned; externally peer reviewed.

**Data availability statement** No data are available. All data and materials relating to this research are from the Haematological Malignancy Research Network and are archived and maintained by the first and last author, according to organisational and ethical regulations. Data are not publicly available due to the risk of participant

identification from specific contexts revealed when reading entire transcripts and due to the terms and conditions regarding the release of data to third parties upon which ethical approvals for this study were contingent. Reasonable requests for further information relating to this data can be made to the corresponding author.

**ORCID iDs**
Dorothy McCaughan http://orcid.org/0000-0001-5388-2455
Eve Roman http://orcid.org/0000-0001-7603-3704
Alexandra Smith http://orcid.org/0000-0002-1111-966X
Debra Howell http://orcid.org/0000-0002-7521-7402

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
