## [Reviewer comments · BMJ Open]

ARTICLE DETAILS

TITLE (PROVISIONAL)	Treatment decision making (TDM): A qualitative study exploring the perspectives of patients with chronic haematological cancers
AUTHORS	McCaughan, Dorothy; Roman, Eve; Smith, Alexandra; Patmore, Russell; Howell, Debra

VERSION 1 – REVIEW

REVIEWER	Campbell, Karen Edinburgh Napier University, School of health and Social Care
REVIEW RETURNED	20-Aug-2021

GENERAL COMMENTS	Thank you for allowing me to review your script - please find my comments below. Treatment decision making (TDM): a qualitative study exploring the perspectives of patients with chronic haematological cancers. Abstract: represents what is written in the script. Background: supports the study and the objectives. Methods: The interview schedule appears to be more information orientated than decision making. I do understand that the interview may have required to be assessed first as it underscores decision making – I wonder if this paper could explore more the relationships/connections between the decision making and the information. It is only in observation that the framework afforded by the interview schedule, which is precise for a semi structured interviews, may have been a better reporting system than a general generation of themes. Sample and setting: the study states it is a UK HMRN , details to which regions the patients came from in the UK, would benefit the reader to know the distribution across the UK. Sampling strategy gives an adequate variety of disease types, age and time from diagnosis. It is noted from the table 1 that of the 35 participants 28 were currently under observation and 11 of the 28 were under observation and had had no other treatment options at the time of interview. This is not reported as having an influence on the data and it would be important to acknowledge that this in fact could change the relationship with the clinician and or perspective of future treatment decisions. Were the 11 patients given options of watch and wait Vs treatment? How often were these participants seeing the clinicians? Data collection: line 34 it may benefit the reader to know what type of clinician inputted to the interview schedule. Data Analysis: as above I feel the analysis would have benefited from a framework rather than themes generated, but that is an observation.
--

Ethical considerations: have been addressed for this study.
 Patient and public involvement: show due consideration for the overall process.

Results: It would be interesting for the reader to know more about the interviews which were attended by another person. Even if this was not the overall intention to do this at the start of the study, how were the extra participant data used in the analysis and was the interview different than the single interviews. It may be more appropriate to put the information that this inclusion added value to the prompting of the responses from the participants, as well as the limitations section. I feel that it would add to the script if you detailed the numbers of participants on each pathway line 48 49 page 7.

The table is appreciated as it does allow review of the participants and their pathway. The table is presented from P1 to P35 I am assuming that the interviews were done in that order, hence the layout. However, in analysing the specific detail of the table it may be better to group the participants in order of number of treatments to support the relationship with the quotes and the type of pathway and the participant response to the TDM process. Or the brackets after the quotes contains the full gender, treatment and age and disease as the reader then can quickly assimilate the information without going back and forward to the table.

The themes looking at the interview schedule I would have expected one on 'information' received and the influence on TDM, it may benefit the reader to know how the interview schedule changed to reflect the concurrent analysis over time.

Theme 1:
 Line 13/14 page 9 you state that most participants welcomed being informed – it would be good to understand here that if most were on an observation pathway, how often their interaction was (if possible). What were the touch points that required TDM?
 There is a variation in the type of participant and their pathway for the sections on page 9 but this is not reflected in the supporting findings. Could this be commented upon as reflecting the whole group and /or supply one quote to support the findings or two different quotes to reflect the negative and positive cases. Later in the script you state that the TDM is more extensive for the MM patients. Could this be reflected here?

Quotes on Page 10
 From looking at the quotes and matching with Table 1 – P9 is stating that they were frightened “you will come out of it .. I am so certain of that” however, the table would suggest that they are on a watch and wait pathway.
 Feels like there are more elements of depth required for this section line 40 page 10. I think the table could let the reader know the hospital setting in which they are being treated – As in is it a specialist haematologist in a cancer centre or a haematologist that works across all the haematology diseases and therefore does not appear as a speciality in one specific area.
 Looking at the quotes at the bottom of page 10 – is it reassurance rather than trust.

Theme 2: Factors implicated in patient involvement
 Theme 5 seems to be incorporated in themes 2 – line 44/45 page 11. Consider merging these two themes.
 This section will look very different from the others as it will be just quoting and a diagram. Might it look better if there is a paragraph at least introducing the subtheme development[text].

	Themes 3: Perceptions Quoting on P6 appears to contradict – as they say they did intelligent searches and then later say you reach the buffers at some point - it is too difficult to understand. You do acknowledge in your limitations the representation of the population sample – however this would suggest a very high level of knowledge and access to knowledge. How does that reflect in your analysis with the non – engagers? The quotes would suggest that you did analyses the relative’s responses to the interview schedule. How is this reflected in the Title and design of the study? Page 14 – line 32 in brackets – you use ‘in denial’ is this the patient’s words? Several of the quotations for the paragraph beginning factors line 28/29 page 14 is a repeat of the information access, in theme 2. For me this section is in stark contrast to the well informed population above. Consider a few quotes that would give a best fit to the findings. Theme 4: Paragraph two would suggest that they entered the clinic to be advised to take treatment and they had it there and then – it may just need a little clarity if that was the context. Or why was this so urgent that they could not take time to process the information and discuss with family? This would suggest that there is a temporal aspect to the TDM negating involvement in the process. Theme 5: there is an overlap with theme 2. In this theme the quotation is in a box – I would suggest this is for all or none in the script. Again, there are a few quotes which do not fit with the findings description which may require fuller explanation. Is there a one or two that could represent the group. There is not a diagram for all the themes just theme 2 and 3 consider the balance of the article. Discussion: The findings section and the diagrams do not display interconnected mechanisms at play in this research, but I do agree that they give the reader insight into the TDM. To give an interconnected display would require a diagram for all 5 themes and where they possibly overlap. Page 19-line 11/12 your finding shows a contrast between the educated and the uneducated and the access to information, consider reflecting this in the discussion at this point. In the discussion you state that Myeloma participants appear to find this the greatest – is that a reflection on the amount and types of treatment rather than a watch and wait pathway? Your findings may need to reflect the weight that you place on this in the discussion by providing a contrast. In your discussion you state that they struggle to weigh up the risks, adverse quality of life and prognostic uncertainty – is this reflected in your findings/quotations? Line 32 – the argument flow in this paragraph seems to not be attached to the previous paragraph or the one below. Consider removing as this is clinician TDM not patient TDM. Page 20 line 19 the distinction between age, educated level and highly motivated is not explored in the findings. Consider updating the findings to reflect the discussion. Next paragraph – is it that they didn’t want to engage, or do they just not understand the complexity – it may not always be about health literacy because it is a very complex science and TDM. Agree with Line 38/39. Line 58 and on progression of disease.
--	---

	Clinical Implications: The clinical implications section is weak and should give recommendation that would have clinical utility e.g. underpinned by an assessment overtime of the patient in order to provide the most optimum conditions for the transfer of information. In summary the results section required reviewed. Multiple quotes do give a sense of the variety of participant response but may not always be supported by the wrap around writing and at time can look conflicting in nature. The thematic representation of the results requires review.
--	---

VERSION 1 – AUTHOR RESPONSE

Reviewer: Dr. Karen Campbell, Edinburgh Napier University

COMMENT: Abstract: represents what is written in the script.

RESPONSE: No change made

COMMENT: Background: supports the study and the objectives.

RESPONSE: No change made

COMMENT: Methods: The interview schedule appears to be more information orientated than decision making. I do understand that the interview may have required to be assessed first as it underscores decision making – I wonder if this paper could explore more the relationships/connections between the decision making and the information.

RESPONSE: Thank you for your comment. We agree that the information patients receive/seek out (or do not receive/do not seek out) is closely linked to their potential for playing a role in treatment decision making (TDM). However, our findings suggest that information (or lack thereof) is only one of a number of factors that impact on patients' willingness/ability to become involved in TDM, as described in Theme 2, pages 12-13 of the Manuscript, and in Figure 1. Our aim in this Manuscript was, primarily, to provide an overview of the range of factors seemingly implicated in TDM, as indicated to us by our study sample. Preparation of a further paper, with a central focus on the significance of information to patients, is currently underway.

COMMENT: It is only in observation that the framework afforded by the interview schedule, which is precise for a semi structured interviews, may have been a better reporting system than a general generation of themes.

RESPONSE: As indicated on page 6 of the Manuscript, lines 26-27, the interview topic guide was used flexibly so as to allow patients to 'tell their story' and interviewees were encouraged to expand on any and all issues related to treatment decision making that were of importance to them. Accordingly, individual interviewees' responses did not always easily 'slot into' an 'answer section' for individual questions in the topic guide. This could have posed problems if the topic guide had been used as a framework for analysis, potentially constricting and circumscribing findings. We therefore elected to use Sandelowski's (2000) well-established approach of thematic content analysis, as more suited to the nature of the data obtained.

Reference: Sandelowski M. 2000. Focus on research methods: Whatever happened to qualitative description? *Res Nurs Health*. 23(4):334-340. Available from: [https://doi.org/10.1002/1098-240x\(200008\)23:4<334::aid-nur9>3.0.co;2-g](https://doi.org/10.1002/1098-240x(200008)23:4<334::aid-nur9>3.0.co;2-g).

COMMENT: Sample and setting: the study states it is a UK HMRN , details to which regions the patients came from in the UK, would benefit the reader to know the distribution across the UK.

RESPONSE: Thank you for this helpful comment. We have amended the Manuscript (page 5, section 2.2, lines 25-26) to include the sentence 'HMRN includes a population of around 4 million people in Yorkshire and Humberside, and has a similar age and sex profile to the UK and a comparable socio-economic and urban/rural distribution.'

COMMENT: Sampling strategy gives an adequate variety of disease types, age and time from diagnosis. It is noted from the table 1 that of the 35 participants 28 were currently under observation and 11 of the 28 were under observation and had had no other treatment options at the time of interview. This is not reported as having an influence on the data and it would be important to acknowledge that this in fact could change the relationship with the clinician and or perspective of future treatment decisions. Were the 11 patients given options of watch and wait Vs treatment? How often were these participants seeing the clinicians?

RESPONSE: Thank you for your comment. We have updated Table 1 based on further pathway data collected routinely within HMRN and it now shows that seven patients began/remained on observation, and had not received any treatment prior to interview. You are correct that these patients may have had less frequent visits and contact with haematology HCPs than those attending consultations to discuss/receive treatment. We have therefore amended the Manuscript to reflect these points. Please see page 20 (lines 21-24), where we have inserted the sentence: 'Dissatisfaction amongst this group of patients was possibly, in part, attributable to the fact that patients with CLL often had infrequent contact with haematology HCPs during long periods of observation, with months sometimes elapsing between appointments.'

COMMENT: Data collection: line 34 it may benefit the reader to know what type of clinician inputted to the interview schedule.

RESPONSE: We have amended the Manuscript (page 6, line 23), which now reads: 'based on research literature and input from clinicians (haematology specialist consultants and nurses) and piloted with 2 patients...'

COMMENT: Data Analysis: as above I feel the analysis would have benefited from a framework rather than themes generated, but that is an observation.

RESPONSE: We have already responded to this comment – please see above.

COMMENT: Ethical considerations: have been addressed for this study.

RESPONSE: Thank you

COMMENT: Patient and public involvement: show due consideration for the overall process.

RESPONSE: Thank you

COMMENT: Results: It would be interesting for the reader to know more about the interviews which were attended by another person. Even if this was not the overall intention to do this at the start of the study, how were the extra participant data used in the analysis and was the interview different than the single interviews. It may be more appropriate to put the information that this inclusion added value to the prompting of the responses from the participants, as well as the limitations section.

RESPONSE: Thank you for these helpful comments. As you suggest, we have included new text in the Strengths and Limitations section to underscore the value of involvement of relatives in the study. To this end, the Manuscript (page 23, lines 6-8) now includes the sentence: 'Relatives' participation enhanced the quality of the data collected, through contribution of their own perspectives, prompting patients, and, on occasion, corroboration and/or clarification of patients' accounts.'

COMMENT: I feel that it would add to the script if you detailed the numbers of participants on each pathway line 48 49 page 7.

RESPONSE: Thank you for your comment. We have now updated Table 1 and have inserted the numbers of participants on each pathway into the Manuscript text (see page 7, line 31-page 8, line 1).

COMMENT: The table is appreciated as it does allow review of the participants and their pathway. The table is presented from P1 to P35 I am assuming that the interviews were done in that order, hence the layout. However, in analysing the specific detail of the table it may be better to group the participants in order of number of treatments to support the relationship with the quotes and the type of pathway and the participant response to the TDM process. Or the brackets after the quotes contains the full gender, treatment and age and disease as the reader then can quickly assimilate the information without going back and forward to the table.

RESPONSE: We have given this comment full consideration. Our view is that the table has been provided for the purpose of reference, allowing the reader to draw a link between participant characteristics and the study findings, and that the table is clearly presented. We have now inserted into the Manuscript text numbers of patients on different pathways (see Manuscript page 7, line 31-page 8, line1). Table 1 lists interviewees in the order in which they were interviewed, promoting transparency of the analytical process, by indicating the types of haematological diagnoses the interviewer had encountered at any one point in time.

COMMENT: The themes looking at the interview schedule I would have expected one on 'information' received and the influence on TDM

RESPONSE: This comment is similar to one made above, and we reproduce here our earlier response: 'Thank you for your comment. We agree that the information patients receive/seek out (or do not receive/ do not seek out) is closely linked to their potential for playing a role in treatment decision making (TDM). However, our findings suggest that information (or lack thereof) was only one of a number of factors that impact on patients' willingness/ability to become involved in TDM, as described in Theme 2, pages 12-13 of the Manuscript, and in Figure 1. Our aim in this Manuscript was, primarily, to provide an overview of the range of factors seemingly implicated in TDM, as indicated to us by our study sample. Preparation of a further paper, with a central focus on the significance of information to patients, is currently underway.'

COMMENT: It may benefit the reader to know how the interview schedule changed to reflect the concurrent analysis over time.

RESPONSE: As we indicated on Manuscript Page 6, lines 24-26, the topic schedule was modified as data collection/analysis progressed. Modification mainly took the form of adding in new prompts, rather than new questions. New prompts were linked to what participants told us, and included in order to promote exploration of particular aspects of views/experiences related to specific questions in subsequent interviews. The topic guide that is appended for submission of the Manuscript represents its final 'version'.

COMMENT: Theme 1: Line 13/14 page 9 you state that most participants welcomed being informed – it would be good to understand here that if most were on an observation pathway, how often their interaction was (if possible). What were the touch points that required TDM?

RESPONSE: We are unable to provide exact details of how often patients met with clinicians for each of the study participants. Episodes of interaction varied in frequency over time, depending on the patient's diagnosis, disease progression, treatment and so on, and it is not feasible to provide this level of detail for each individual, which would be better suited to a quantitative study/analysis of Health Episode Statistics (HES). Moreover, we relied on what patients told us about the frequency of clinical encounters/interactions, and did not attempt to 'check' what participants had said during interview against HMRN data or other clinical records. Patients' accounts were (unsurprisingly) often somewhat vague concerning whether they were on a 1 monthly, 3 monthly or 6 monthly monitoring

regimen at any one time point. 'Touchpoints' which triggered TDM related to clinical signs and symptoms of disease progression or relapse, and to clarify this we inserted the following text (Manuscript page 4): 'These pathways are associated with variations in the extent of hospital activity, with periods of disease progression, relapse and treatment requiring more consultations and decisions, and W&W having fewer contact points, often months apart.

COMMENT: There is a variation in the type of participant and their pathway for the sections on page 9 but this is not reflected in the supporting findings. Could this be commented upon as reflecting the whole group and /or supply one quote to support the findings or two different quotes to reflect the negative and positive cases.

RESPONSE: In the Manuscript pages 7-8, it is stated that: 'Prior to interview, patients had experienced different treatment pathways, according to diagnosis and disease progression; some (7) had started and remained on W&W, others (22) had started treatment, and a further group (6) had experienced multiple lines of chemotherapy before progressing to stem cell transplant. Patient characteristics and individuals' treatment pathways, ascertained from HMRN routine data collection, and patient self-report, can be found in Table 1.' Unless specified otherwise, findings should be understood to be applicable to the whole group of participants included in the study.

COMMENT: Later in the script you state that the TDM is more extensive for the MM patients. Could this be reflected here?

RESPONSE: As the reviewer says, we describe TDM in patients diagnosed with myeloma in some detail later in the Manuscript, and feel it would be inappropriate to include it within this section of the text, which is making more general points.

COMMENT: Quotes on Page 10 From looking at the quotes and matching with Table 1 – P9 is stating that they were frightened "you will come out of it .. I am so certain of that" however, the table would suggest that they are on a watch and wait pathway.

RESPONSE: Thank you for highlighting this issue, which has been resolved by updating Table 1.

COMMENT: Feels like there are more elements of depth required for this section line 40 page 10. I think the table could let the reader know the hospital setting in which they are being treated – As in is it a specialist haematologist in a cancer centre or a haematologist that works across all the haematology diseases and therefore does not appear as a speciality in one specific area.

RESPONSE: Thank you for your comment. In the Manuscript, the text included in pages 10-12 provides a resume of some patients' perceptions concerning their views of the level of expertise and degree of specialism of the HCPs looking after them; their comments indicate how these perceptions influenced patients' trust in clinicians making treatment decisions on patients' behalf. We did not attempt to establish the actual place of treatment, or to provide any details concerning individual clinician(s), for three reasons: (1) this would be almost impossible to establish with accuracy, as clinicians and patients frequently move(d) across care settings; (2) from an ethics perspective, we wished to ensure anonymity (participant/place/health care practitioner); (3) our aim was to draw inferences from broader patient perspectives, rather than provide a detailed account of the 'actual facts' of each individual's specific context.

COMMENT: Looking at the quotes at the bottom of page 10 – is it reassurance rather than trust.

RESPONSE: Thank you for this comment. We would suggest that reassurance and trust are semantically related, and that reassurance is an elemental component of trust. Pearson and Raeke (2000) have examined the construct of trust and suggest that 'Patient trust is a complicated, multidimensional construct which has been described in many ways.' These authors write that 'Some theorists consider patient trust to be a set of beliefs or expectations that a physician will behave in a

certain way. Others have stressed a more affective nature of trust, identifying patient trust as a reassuring feeling of confidence or reliance in the physician and the physician's intent.'

Reference: Pearson SD, Raeka LH. Patients' Trust in Physicians: Many Theories, Few Measures and Little Data. J Gen Intern Med 2000; 15: 509-513.

COMMENT: Theme 2: Factors implicated in patient involvement Theme 5 seems to be incorporated in themes 2 – line 44/45 page 11. Consider merging these two themes.

RESPONSE: Thank you for your comment. We do mention 'level of support from others' in Theme 2, as a factor implicated in treatment decision making, while Theme 5 is solely focussed on the level of support from others. Based on your comment, we have introduced a phrase in Theme 2 (page 12, line 22), which now reads: 'and the level of support from others (elaborated in Theme 5).'

COMMENT: This section will look very different from the others as it will be just quoting and a diagram. Might it look better if there is a paragraph at least introducing the subtheme development [text]

RESPONSE: Thank you for this comment. We have now added into the text of the Manuscript, page 12, lines 23-24), the following text: 'These factors were drawn together to develop this theme and are summarised in Figure 1, with quotes (below) illustrating each component.'

COMMENT: Themes 3: Perceptions

Quoting on P6 appears to contradict – as they say they did intelligent searches and then later say you reach the buffers at some point - it is too difficult to understand. You do acknowledge in your limitations the representation of the population sample – however this would suggest a very high level of knowledge and access to knowledge. How does that reflect in your analysis with the non – engagers?

RESPONSE: Thank you for your comment. For purposes of clarification we have now added the following sentence to the Manuscript, page 14, lines 15-19: 'Even these patients, who were well equipped to retrieve and assimilate information, reached a limit to their capacity to understand information relating to their haematological cancer at a specific juncture; other study participants indicated that they had difficulty understanding much/most of the information encountered.'

COMMENT: The quotes would suggest that you did analyses the relative's responses to the interview schedule. How is this reflected in the Title and design of the study?

RESPONSE: Our study was designed to recruit patients with chronic haematological diseases; we did not specifically aim to recruit patients' relatives, although if relatives were present at the time of interview, they were able to participate if they so wished, and the patient agreed. In some cases relatives contributed very little to the interview; in a minority of cases, they took a more active part. Although data from relatives were analysed alongside those from patients, the vast majority of quotations cited in the paper are from the targeted patient group. Given this, we do not feel that the relatively small amount of data from relatives justifies a change in the title of the Manuscript. However, at the reviewer's suggestion we have included the following sentence in the section on Strengths and Limitations (page 23, lines 6-8): 'Relatives' participation enhanced the quality of the data collected, through contribution of their own perspectives, prompting patients, and, on occasion, corroboration and/or clarification of patients' accounts.'

COMMENT: Page 14 – line 32 in brackets – you use 'in denial' is this the patient's words?

RESPONSE: Yes, Patient 35 referred to be being 'in denial' and these words have now been attributed to her, with the P35 identifier attached – see Manuscript page 15, line 22).

COMMENT: Several of the quotations for the paragraph beginning factors line 28/29 page 14 is a repeat of the information access, in theme 2. For me this section is in stark contrast to the well informed population above. Consider a few quotes that would give a best fit to the findings.

RESPONSE: We have checked the quotations in the paragraph beginning 'Factors...' (now on page 15 of the amended Manuscript) and those in Theme 2, and they do not appear to overlap or 'repeat'. The different quotations in each section were drawn from a range of individuals (no two quotes are from the same patient), and they underscore the difficulties experienced by many participants with regard to access/understanding information, as well as the varied nature of difficulties experienced e.g. clinicians using complex language rather than 'layman's terms'; a patient who described himself as 'not computer literate'; lack of understanding of significance of results from blood tests; attitude to internet, seen as 'annoying'; unable to retain information due to 'fuzzy' memory, and so on. We have emphasised that many patients experienced difficulties with information in response to an earlier comment from the reviewer, and have inserted into the Manuscript text (page 14, lines 17-18): 'other study participants indicated they had difficulty understanding much/most of the information they encountered'.

COMMENT: Theme 4: Paragraph two would suggest that they entered the clinic to be advised to take treatment and they had it there and then – it may just need a little clarity if that was the context. Or why was this so urgent that they could not take time to process the information and discuss with family? This would suggest that there is a temporal aspect to the TDM negating involvement in the process.

RESPONSE: Thank you for your comment which highlights the need for clarification. Some of the patients with CLL referred to the short interval (in some cases, 2 weeks) between being told they would need treatment, and treatment commencing. We have revised this paragraph and it now reads: 'Some patients on W&W, who subsequently went on to require treatment, reported little or no involvement in TDM about the type of therapy to be given, as there was only a single relevant option. In this context the decision was said to be 'automatic' and that it would be the 'standard' treatment. Some of these participants said they would have welcomed having more time to discuss the proposed treatment with family members, and to consider whether or not to accept it; a few recalled their consultant strongly recommending that they accept the treatment offered.' (Manuscript page 16, lines 22-27)

COMMENT: Theme 5: there is an overlap with theme 2.

RESPONSE: As noted previously, we do mention 'level of support from others' in Theme 2, as a factor implicated in treatment decision making, while Theme 5 is solely focussed on the level of support from others. Based on your comment, we have introduced a phrase in Theme 2, page 12, line 22, which now reads: 'and the level of support from others (elaborated in Theme 5).'

COMMENT: In this theme the quotation is in a box – I would suggest this is for all or none in the script. Again, there are a few quotes which do not fit with the findings description which may require fuller explanation. Is there a one or two that could represent the group.

RESPONSE: Thank you for these comments. We have now removed the box containing quotations, and instead have woven quotations into the body of the text. As suggested, we have also changed some quotations to match/illustrate what is said in the text more closely. Please see Manuscript pages 18-19) for these amendments, which we believe have improved the section considerably.

COMMENT: There is not a diagram for all the themes just theme 2 and 3 consider the balance of the article.

RESPONSE: The reviewer is raising a stylistic point here concerning the 'balance of the article.' We incorporated diagrams to clarify more complex material, while we felt the content of some themes could be readily digested from what was said in the text, without the need of a supporting diagram.

Discussion:

COMMENT: The findings section and the diagrams do not display interconnected mechanisms at play in this research, but I do agree that they give the reader insight into the TDM. To give an interconnected display would require a diagram for all 5 themes and where they possibly overlap.

RESPONSE: Thank you for your comment. We have used diagrams (Figures 1 and 2) to promote and enhance understanding of the more complex material included in the Manuscript, as presented in Theme 2 and Theme 3, but we felt that textual descriptions sufficed for Themes 1, 4 and 5. We did not set out to produce an overall, inter-connecting diagram, that might act as a conceptual or theoretical framework, or lens, for examining TDM, as a programme of research to provide an interconnected, diagrammatic overview of TDM already exists (see: Waldron T, Carr T, McMullan L, et al. Development of a programme theory for shared decision making: a realist synthesis. BMC Health Services Research. 2020; 20:59. Available from: <https://doi.org/10.1186/s12913-019-4649-1>).

COMMENT: Page 19-line 11/12 your finding shows a contrast between the educated and the uneducated and the access to information, consider reflecting this in the discussion at this point.

RESPONSE: We have considered this comment carefully; we deliberately have not drawn any inferences from our findings based on patients' level of education. Neither Figure 1 ('Factors Implicated in Patient Involvement in Treatment Decisions') nor Figure 2 ('Continuum of Characteristics Associated with Proactive and Non-Proactive Approaches to Involvement in Treatment Decisions') refer to level of education. We suggest that focusing on education alone would oversimplify the issue as this is not the sole predictor of information access or understanding. For example, knowledge/education in one area does not necessarily translate to another; and the desire to seek-out information is not the sole domain of the highly educated. In the Manuscript (page 21) we noted (for descriptive purposes) that the 5 seemingly proactive information-seeking patients amongst the sample shared certain characteristics (male, mainly younger, educated to degree level, and highly motivated). However, we have not made any general inferences based on these data, or any other data concerning the level of education of any of the study participants.

COMMENT: In the discussion you state that Myeloma participants appear to find this the greatest – is that a reflection on the amount and types of treatment rather than a watch and wait pathway? Your findings may need to reflect the weight that you place on this in the discussion by providing a contrast.

RESPONSE: We have amended the text (Manuscript page 20, lines 13-14) to now read: 'Participants with myeloma, who had most decisions to make, due to the nature of their cancer...'

COMMENT: In your discussion you state that they struggle to weigh up the risks, adverse quality of life and prognostic uncertainty – is this reflected in your findings/quotations?

RESPONSE: This point in the discussion concerning patients with myeloma reflects the findings as reported in Manuscript pages 17-18, reproduced below:

'Disease progression resulted in some patients with myeloma feeling overwhelmed when faced with difficult treatment decisions, and unable to choose between options. Factors compounding this included the intensive nature of proposed treatments (such as stem cell transplant) and their impact on quality of life; the limited "returns" that some treatments seemed to offer, compared to the consequences of associated risks, such as infection; and the uncertainty and unpredictability of outcomes.

'Honestly, my head is exploding with all this...it's just like a big crushing thing to me...I think I am quite strong but this is doing me in' (P35)

'the big one [decision] was the second stem cell transplant...I was really struggling to make the decision as to whether I wanted to go for it' (P18)

'they [doctors] sort of said, well the average remission after the stem cell transplant is, I think either

12-24 months, or 18-24 months, something like that, and there was I thinking, right I'm going for the 10-year option, so that was quite a shock' (P18)

'Myeloma is a very individual disease. You get the same treatment, same, same this, same that but you have different outcomes and things.' (P28)

COMMENT: Line 32 – the argument flow in this paragraph seems to not be attached to the previous paragraph or the one below. Consider removing as this is clinician TDM not patient TDM.

RESPONSE: We are puzzled by this comment, as there is a link to the statement in the previous paragraph, that patients in the study with CLL 'expressed dissatisfaction that they had not been given the opportunity to fully consider, and accept, or decline, treatment' and the following paragraph (which the reviewer refers to), highlighting physicians' perspectives, that where there is good evidence for the selection of a specific treatment, then 'In such instances, the decision is not one of selecting between options, but rather whether the patient chooses to accept or decline treatment.' We suggest that one of the purposes of the discussion is to draw in opinions from other important stakeholders, with reference to the broader research literature and evidence from other studies.

COMMENT: Page 20 line 19 the distinction between age, educated level and highly motivated is not explored in the findings. Consider updating the findings to reflect the discussion.

RESPONSE: Thank you for your comment. To clarify, our study was not designed to conduct in-depth exploration of inter-relationships between specific factors such as age, educational attainment, and motivation; the small-scale, purposive, qualitative sample employed in our study would not enable us to imply that such relationships exist, or make generalisations based on the findings. We note, however, for descriptive purposes, that the patients whose information seeking activities might be regarded as proactive (see Manuscript Figure 2, as well as Theme 3), who were in their 50s and early 60s, were comparatively younger than the majority of patients included in the study (median age 67 years).

COMMENT: Next paragraph – is it that they didn't want to engage, or do they just not understand the complexity – it may not always be about health literacy because it is a very complex science and TDM. Agree with Line 38/39. Line 58 and on progression of disease.

RESPONSE: Thank you for your comment. As the reviewer notes, many patients did not want to be involved in TDM, and were happy to trust clinicians to make decisions on their behalf, while others indicated that they might have adopted a more active role in TDM, but perceived the complex nature of their haematological cancer as a deterrent.

COMMENT: Clinical Implications: The clinical implications section is weak and should give recommendation that would have clinical utility e.g. underpinned by an assessment overtime of the patient in order to provide the most optimum conditions for the transfer of information.

RESPONSE: Thank you for your comment. We agree that assessment over time of the patient is of central importance and thus we had already included in the Manuscript (page 23, lines 23-24) a sentence to that effect: 'Our findings suggest that our interviewees varied in their preference for involvement in TDM according to intrinsic, contextual, and disease-related factors, requiring clinicians to assess individuals' preferences for engagement at multiple time points.' In order to further underscore this recommendation, we have expanded this sentence: 'Our findings suggest that our interviewees varied in their preference for involvement in TDM according to intrinsic, contextual, and disease-related factors, requiring clinicians to assess individuals' preferences for engagement at multiple time points over the course of their haematological cancer pathway.' (Manuscript page 23, lines 23-24).

COMMENT: In summary the results section required reviewed. Multiple quotes do give a sense of the variety of participant response but may not always be supported by the wrap around writing and at time can look conflicting in nature. The thematic representation of the results requires review.

RESPONSE: Thank you for this final comment which refers to issues raised elsewhere which we hope we have addressed/resolved through our responses and amendments to the Manuscript.

VERSION 2 – REVIEW

REVIEWER	Campbell, Karen Edinburgh Napier University, School of health and Social Care
REVIEW RETURNED	20-Oct-2021

GENERAL COMMENTS	Thank you for allowing me to review your script again. You have mostly answered my comments. The comments where I am asking for more description or stating a different way of looking at the data is only because at point this is not transparent for the reader. Now that you have highlighted that this is one of two papers it makes sense where you have made the distinction between TDM and information, but it is also the difficulty often that a paper feels like it is missing a part. Treatment decision making: A qualitative study exploring the perspectives of patients with chronic haematological cancers: Response to reviewer’s comments Reviewer: Dr. Karen Campbell, Edinburgh Napier University COMMENT: Abstract: represents what is written in the script. RESPONSE: No change made COMMENT: Background: supports the study and the objectives. RESPONSE: No change made COMMENT: Methods: The interview schedule appears to be more information orientated than decision making. I do understand that the interview may have required to be assessed first as it underscores decision making – I wonder if this paper could explore more the relationships/connections between the decision making and the information. RESPONSE: Thank you for your comment. We agree that the information patients receive/seek out (or do not receive/do not seek out) is closely linked to their potential for playing a role in treatment decision making (TDM). However, our findings suggest that information (or lack thereof) is only one of a number of factors that impact on patients’ willingness/ability to become involved in TDM, as described in Theme 2, pages 12-13 of the Manuscript, and in Figure 1. Our aim in this Manuscript was, primarily, to provide an overview of the range of factors seemingly implicated in TDM, as indicated to us by our study sample. Preparation of a further paper, with a central focus on the significance of information to patients, is currently underway. Response: I appreciate that you are working up another paper on the topic – I think to focus the reader then a statement on the last sentence in last paragraph could just be tweaked. Pg 5 last paragraph in which you identify the aim of the study – starting this study (last sentence) I would change the emphasis to providean evidence base – then say another strand will be published separately on information. Just makes the reader focus.
--

	COMMENT: It is only in observation that the framework afforded by the interview schedule, which is precise for a semi structured interviews, may have been a better reporting system than a general generation of themes. RESPONSE: As indicated on page 6 of the Manuscript, lines 26-27, the interview topic guide was used flexibly so as to allow patients to 'tell their story' and interviewees were encouraged to expand on any and all issues related to treatment decision making that were of importance to them. Accordingly, individual interviewees' responses did not always easily 'slot into' an 'answer section' for individual questions in the topic guide. This could have posed problems if the topic guide had been used as a framework for analysis, potentially constricting and circumscribing findings. We therefore elected to use Sandalowski's (2000) well-established approach of thematic content analysis, as more suited to the nature of the data obtained. Reference: Sandelowski M. 2000. Focus on research methods: Whatever happened to qualitative description? Res Nurs Health. 23(4):334-340. Available from: https://doi.org/10.1002/1098-240x(200008)23:43.0.co;2-g. Response: In your response I can see that the difficulty is that you are reporting one part of a study here and therefore when reading the paper, it does feel at times there is a missing link. Again, I think this is about stating at the end of the background section or in findings that this is only reporting a strand of the study. Why not just thematic analysis? COMMENT: Sample and setting: the study states it is a UK HMRN , details to which regions the patients came from in the UK, would benefit the reader to know the distribution across the UK. RESPONSE: Thank you for this helpful comment. We have amended the Manuscript (page 5, section 2.2, lines 25-26) to include the sentence 'HMRN includes a population of around 4 million people in Yorkshire and Humberside, and has a similar age and sex profile to the UK and a comparable socioeconomic and urban/rural distribution.' Response: This reads better now. Would be interesting to know how well served the services are in providing sub type specific services and clinical trial access. All things that may change the decision making. This may be in your Information paper? COMMENT: Sampling strategy gives an adequate variety of disease types, age and time from diagnosis. It is noted from the table 1 that of the 35 participants 28 were currently under observation and 11 of the 28 were under observation and had had no other treatment options at the time of interview. This is not reported as having an influence on the data and it would be important to acknowledge that this in fact could change the relationship with the clinician and or perspective 2 of future treatment decisions. Were the 11 patients given options of watch and wait Vs treatment? How often were these participants seeing the clinicians? RESPONSE: Thank you for your comment. We have updated Table 1 based on further pathway data collected routinely within HMRN and it now shows that seven patients began/remained on observation, and had not received any treatment prior to interview. You are correct that these patients may have had less frequent visits and contact with haematology HCPs than those attending consultations to discuss/receive treatment. We have therefore amended the Manuscript to reflect these points. Please see page 20 (lines 21-24), where we have inserted the sentence: 'Dissatisfaction amongst this group of patients was possibly, in
--	--

part, attributable to the fact that patients with CLL often had infrequent contact with haematology HCPs during long periods of observation, with months sometimes elapsing between appointments.'

Response: That reads better

COMMENT: Data collection: line 34 it may benefit the reader to know w what type of clinician inputted to the interview schedule.

RESPONSE: We have amended the Manuscript (page 6, line 23), which now reads: 'based on research literature and input from clinicians (haematology specialist consultants and nurses) and piloted with 2 patients...'

COMMENT: Data Analysis: as above I feel the analysis would have benefited from a framework rather than themes generated, but that is an observation.

RESPONSE: We have already responded to this comment – please see above.

COMMENT: Ethical considerations: have been addressed for this study.

RESPONSE: Thank you

COMMENT: Patient and public involvement: show due consideration for the overall process.

RESPONSE: Thank you

COMMENT: Results: It would be interesting for the reader to know more about the interviews which were attended by another person. Even if this was not the overall intention to do this at the start of the study, how were the extra participant data used in the analysis and was the interview different than the single interviews. It may be more appropriate to put the information that this inclusion added value to the prompting of the responses from the participants, as well as the limitations section.

RESPONSE: Thank you for these helpful comments. As you suggest, we have included new text in the Strengths and Limitations section to underscore the value of involvement of relatives in the study. To this end, the Manuscript (page 23, lines 6-8) now includes the sentence: 'Relatives' participation enhanced the quality of the data collected, through contribution of their own perspectives, prompting patients, and, on occasion, corroboration and/or clarification of patients' accounts.'

COMMENT: I feel that it would add to the script if you detailed the numbers of participants on each pathway line 48 49 page 7.

RESPONSE: Thank you for your comment. We have now updated Table 1 and have inserted the numbers of participants on each pathway into the Manuscript text (see page 7, line 31-page 8, line 1).

COMMENT: The table is appreciated as it does allow review of the participants and their pathway. The table is presented from P1 to P35 I am assuming that the interviews were done in that order, hence the layout. However, in analysing the specific detail of the table it may be better to group the participants in order of number of treatments to support the relationship with the quotes and the type of pathway and the participant response to the TDM process. Or the brackets after the 3 quotes contains the full gender, treatment and age and disease as the reader then can quickly assimilate the information without going back and forward to the table.

RESPONSE: We have given this comment full consideration. Our view is that the table has been provided for the purpose of reference, allowing the reader to draw a link between participant characteristics and the study findings, and that the table is clearly presented. We have now inserted into the Manuscript text numbers of patients on different pathways (see Manuscript page 7,

	line 31- page 8, line1). Table 1 lists interviewees in the order in which they were interviewed, promoting transparency of the analytical process, by indicating the types of haematological diagnoses the interviewer had encountered at any one point in time. Response: Thank you for your response, the analysis may point out that regardless of the type of treatment / watch and weight pathway then the general issue is the same regarding the emphasis of the clinician knowing what is best. I think it is a tweak on the emphasis of the paragraph. COMMENT: The themes looking at the interview schedule I would have expected one on 'information' received and the influence on TDM RESPONSE: This comment is similar to one made above, and we reproduce here our earlier response: 'Thank you for your comment. We agree that the information patients receive/seek out (or do not receive/ do not seek out) is closely linked to their potential for playing a role in treatment decision making (TDM). However, our findings suggest that information (or lack thereof) was only one of a number of factors that impact on patients' willingness/ability to become involved in TDM, as described in Theme 2, pages 12-13 of the Manuscript, and in Figure 1. Our aim in this Manuscript was, primarily, to provide an overview of the range of factors seemingly implicated in TDM, as indicated to us by our study sample. Preparation of a further paper, with a central focus on the significance of information to patients, is currently underway.' Response: Again, it would be better if the audience knows that this paper is also coming. COMMENT: It may benefit the reader to know how the interview schedule changed to reflect the concurrent analysis over time. RESPONSE: As we indicated on Manuscript Page 6, lines 24-26, the topic schedule was modified as data collection/analysis progressed. Modification mainly took the form of adding in new prompts, rather than new questions. New prompts were linked to what participants told us, and included in order to promote exploration of particular aspects of views/experiences related to specific questions in subsequent interviews. The topic guide that is appended for submission of the Manuscript represents its final 'version'. COMMENT: Theme 1: Line 13/14 page 9 you state that most participants welcomed being informed – it would be good to understand here that if most were on an observation pathway, how often their interaction was (if possible). What were the touch points that required TDM? RESPONSE: We are unable to provide exact details of how often patients met with clinicians for each of the study participants. Episodes of interaction varied in frequency over time, depending on the patient's diagnosis, disease progression, treatment and so on, and it is not feasible to provide this level of detail for each individual, which would be better suited to a quantitative study/analysis of Health Episode Statistics (HES). Moreover, we relied on what patients told us about the frequency of clinical encounters/interactions, and did not attempt to 'check' what participants had said during interview against HMRN data or other clinical records. Patients' accounts were (unsurprisingly) often somewhat vague concerning whether they were on a 1 monthly, 3 monthly or 6 monthly monitoring regimen at any one time point. 'Touchpoints' which triggered TDM related to clinical signs and symptoms of disease progression or relapse, and to clarify this we
--	--

	inserted the following text (Manuscript page 4): ‘These pathways are associated with variations in the extent of hospital activity, with periods of disease progression, relapse and treatment requiring more consultations and decisions, and W&W having fewer contact points, often months apart. Response: This may be good as a sentence in the limitations. COMMENT: There is a variation in the type of participant and their pathway for the sections on page 9 but this is not reflected in the supporting findings. Could this be commented upon as reflecting the 4 whole group and /or supply one quote to support the findings or two different quotes to reflect the negative and positive cases. RESPONSE: In the Manuscript pages 7-8, it is stated that: ‘Prior to interview, patients had experienced different treatment pathways, according to diagnosis and disease progression; some (7) had started and remained on W&W, others (22) had started treatment, and a further group (6) had experienced multiple lines of chemotherapy before progressing to stem cell transplant. Patient characteristics and individuals’ treatment pathways, ascertained from HMRN routine data collection, and patient self-report, can be found in Table 1.’ Unless specified otherwise, findings should be understood to be applicable to the whole group of participants included in the study. Response : can the brackets be update (n= 7) etc. It looks like a reference number when done this way. COMMENT: Later in the script you state that the TDM is more extensive for the MM patients. Could this be reflected here? RESPONSE: As the reviewer says, we describe TDM in patients diagnosed with myeloma in some detail later in the Manuscript, and feel it would be inappropriate to include it within this section of the text, which is making more general points. COMMENT: Quotes on Page 10 From looking at the quotes and matching with Table 1 – P9 is stating that they were frightened “you will come out of it .. I am so certain of that” however, the table would suggest that they are on a watch and wait pathway. RESPONSE: Thank you for highlighting this issue, which has been resolved by updating Table 1. COMMENT: Feels like there are more elements of depth required for this section line 40 page 10. I think the table could let the reader know the hospital setting in which they are being treated – As in is it a specialist haematologist in a cancer centre or a haematologist that works across all the haematology diseases and therefore does not appear as a speciality in one specific area. RESPONSE: Thank you for your comment. In the Manuscript, the text included in pages 10-12 provides a resume of some patients’ perceptions concerning their views of the level of expertise and degree of specialism of the HCPs looking after them; their comments indicate how these perceptions influenced patients’ trust in clinicians making treatment decisions on patients’ behalf. We did not attempt to establish the actual place of treatment, or to provide any details concerning individual clinician(s), for three reasons: (1) this would be almost impossible to establish with accuracy, as clinicians and patients frequently move(d) across care settings; (2) from an ethics perspective, we wished to ensure anonymity (participant/place/health care practitioner); (3) our aim was to draw inferences from broader patient perspectives, rather than provide a detailed account of the ‘actual facts’ of each individual’s specific context.
--	--

Resources: Thank you for your response – I don't think you should state the actual place of treatment, but Haematology services are so varied and they have different treatment options available even between the four nations that surely this is also a factor. However, I do take your point about protecting the anonymity of both patient and health care staff. I do think that your title covers the this point as in ' perspectives'. Sorry, this brings me around to the general ethos of the qualitative method and its subjective nature – it is how the patient experiences the situation.

COMMENT: Looking at the quotes at the bottom of page 10 – is it reassurance rather than trust. RESPONSE: Thank you for this comment. We would suggest that reassurance and trust are semantically related, and that reassurance is an elemental component of trust. Pearson and Raeke (2000) have examined the construct of trust and suggest that 'Patient trust is a complicated, multidimensional construct which has been described in many ways.' These authors write that 'Some theorists consider patient trust to be a set of beliefs or expectations that a physician will behave in a certain way. Others have stressed a more affective nature of trust, identifying patient trust as a reassuring feeling of confidence or reliance in the physician and the physician's intent.'

Reference: Pearson SD, Raeka LH. Patients' Trust in Physicians: Many Theories, Few Measures and Little Data. J Gen Intern Med 2000; 15: 509-513.

Response: I still think that the quotes you use represent reassurance that it is more than one person making the decision, so it contradicts that they have sole trust in one individual but reassurance that there is more than one brain behind the decision. It may be that it is the wrong quote – or indeed that we interpret the quote differently which is the difficulty of qualitative work and the use of quotes to support our findings in general. I do think there is a wider point which has been raised in studies about being seen in a general haematology clinic rather than a specialist clinic which influences the 'trust' in the decision of treatment.

COMMENT: Theme 2: Factors implicated in patient involvement Theme 5 seems to be incorporated in themes 2 – line 44/45 page 11. Consider merging these two themes. 5

RESPONSE: Thank you for your comment. We do mention 'level of support from others' in Theme 2, as a factor implicated in treatment decision making, while Theme 5 is solely focussed on the level of support from others. Based on your comment, we have introduced a phrase in Theme 2 (page 12, line 22), which now reads: 'and the level of support from others (elaborated in Theme 5).'

COMMENT: This section will look very different from the others as it will be just quoting and a diagram. Might it look better if there is a paragraph at least introducing the subtheme development [text]

RESPONSE: Thank you for this comment. We have now added into the text of the Manuscript, page 12, lines 23-24), the following text: 'These factors were drawn together to develop this theme and are summarised in Figure 1, with quotes (below) illustrating each component.'

COMMENT: Themes 3: Perceptions Quoting on P6 appears to contradict – as they say they did intelligent searches and then later say you reach the buffers at some point - it is too difficult to understand. You do acknowledge in your limitations the representation of the population sample – however this would suggest a very high level of knowledge and access to knowledge. How does that reflect in your analysis with the non – engagers?

RESPONSE: Thank you for your comment. For purposes of clarification we have now added the following sentence to the Manuscript, page 14, lines 15-19: 'Even these patients, who were well equipped to retrieve and assimilate information, reached a limit to their capacity to understand information relating to their haematological cancer at a specific juncture; other study participants indicated that they had difficulty understanding much/most of the information encountered.'

COMMENT: The quotes would suggest that you did analyses the relative's responses to the interview schedule. How is this reflected in the Title and design of the study?

RESPONSE: Our study was designed to recruit patients with chronic haematological diseases; we did not specifically aim to recruit patients' relatives, although if relatives were present at the time of interview, they were able to participate if they so wished, and the patient agreed. In some cases relatives contributed very little to the interview; in a minority of cases, they took a more active part. Although data from relatives were analysed alongside those from patients, the vast majority of quotations cited in the paper are from the targeted patient group. Given this, we do not feel that the relatively small amount of data from relatives justifies a change in the title of the Manuscript. However, at the reviewer's suggestion we have included the following sentence in the section on Strengths and Limitations (page 23, lines 6-8): 'Relatives' participation enhanced the quality of the data collected, through contribution of their own perspectives, prompting patients, and, on occasion, corroboration and/or clarification of patients' accounts.'

COMMENT: Page 14 – line 32 in brackets – you use 'in denial' is this the patient's words?

RESPONSE: Yes, Patient 35 referred to be being 'in denial' and these words have now been attributed to her, with the P35 identifier attached – see Manuscript page 15, line 22).

COMMENT: Several of the quotations for the paragraph beginning factors line 28/29 page 14 is a repeat of the information access, in theme 2. For me this section is in stark contrast to the well informed population above. Consider a few quotes that would give a best fit to the findings.

RESPONSE: We have checked the quotations in the paragraph beginning 'Factors...' (now on page 15 of the amended Manuscript) and those in Theme 2, and they do not appear to overlap or 'repeat'. The different quotations in each section were drawn from a range of individuals (no two quotes are from the same patient), and they underscore the difficulties experienced by many participants with regard to access/understanding information, as well as the varied nature of difficulties experienced e.g. clinicians using complex language rather than 'layman's terms'; a patient who described himself as 'not computer literate'; lack of understanding of significance of results from blood tests; attitude 6 to internet, seen as 'annoying'; unable to retain information due to 'fuzzy' memory, and so on. We have emphasised that many patients experienced difficulties with information in response to an earlier comment from the reviewer, and have inserted into the Manuscript text (page 14, lines 17- 18): 'other study participants indicated they had difficulty understanding much/most of the information they encountered'.

COMMENT: Theme 4: Paragraph two would suggest that they entered the clinic to be advised to take treatment and they had it there and then – it may just need a little clarity if that was the context. Or why was this so urgent that they could not take time to process the information and discuss with family? This would

	suggest that there is a temporal aspect to the TDM negating involvement in the process. RESPONSE: Thank you for your comment which highlights the need for clarification. Some of the patients with CLL referred to the short interval (in some cases, 2 weeks) between being told they would need treatment, and treatment commencing. We have revised this paragraph and it now reads: 'Some patients on W&W, who subsequently went on to require treatment, reported little or no involvement in TDM about the type of therapy to be given, as there was only a single relevant option. In this context the decision was said to be 'automatic' and that it would be the 'standard' treatment. Some of these participants said they would have welcomed having more time to discuss the proposed treatment with family members, and to consider whether or not to accept it; a few recalled their consultant strongly recommending that they accept the treatment offered.' (Manuscript page 16, lines 22-27) COMMENT: Theme 5: there is an overlap with theme 2. RESPONSE: As noted previously, we do mention 'level of support from others' in Theme 2, as a factor implicated in treatment decision making, while Theme 5 is solely focussed on the level of support from others. Based on your comment, we have introduced a phrase in Theme 2, page 12, line 22, which now reads: 'and the level of support from others (elaborated in Theme 5).' COMMENT: In this theme the quotation is in a box – I would suggest this is for all or none in the script. Again, there are a few quotes which do not fit with the findings description which may require fuller explanation. Is there a one or two that could represent the group. RESPONSE: Thank you for these comments. We have now removed the box containing quotations, and instead have woven quotations into the body of the text. As suggested, we have also changed some quotations to match/illustrate what is said in the text more closely. Please see Manuscript pages 18-19) for these amendments, which we believe have improved the section considerably. COMMENT: There is not a diagram for all the themes just theme 2 and 3 consider the balance of the article. RESPONSE: The reviewer is raising a stylistic point here concerning the 'balance of the article.' We incorporated diagrams to clarify more complex material, while we felt the content of some themes could be readily digested from what was said in the text, without the need of a supporting diagram. Response: Personally I still think this will imbalance the article, but am happy with the response. COMMENT: The findings section and the diagrams do not display interconnected mechanisms at play in this research, but I do agree that they give the reader insight into the TDM. To give an interconnected display would require a diagram for all 5 themes and where they possibly overlap. RESPONSE: Thank you for your comment. We have used diagrams (Figures 1 and 2) to promote and enhance understanding of the more complex material included in the Manuscript, as presented in Theme 2 and Theme 3, but we felt that textual descriptions sufficed for Themes 1, 4 and 5. We did not set out to produce an overall, inter-connecting diagram, that might act as a conceptual or theoretical framework, or lens, for examining TDM, as a programme of research to provide an 7 interconnected, diagrammatic overview of TDM already exists (see: Waldron T, Carr T, McMullan L, et al. Development of a programme theory for shared decision making: a realist synthesis.
--	---

BMC Health Services Research. 2020; 20:59. Available from: <https://doi.org/10.1186/s12913-019-4649-1>).

Response: I agree your qualitative methodology was not to develop a theory or an Initial programme theory on TDM. The paper that has been quoted is also a theoretical model about treatment decision making. I can see that you already reference to this in your discussion – I feel it would be interesting to understand the nuances of TDM in haematology patients? Where does your findings overlap or distinct areas to follow up? In relation to this important paper that you have highlighted.

COMMENT: Page 19-line 11/12 your finding shows a contrast between the educated and the uneducated and the access to information, consider reflecting this in the discussion at this point.
RESPONSE: We have considered this comment carefully; we deliberately have not drawn any inferences from our findings based on patients' level of education. Neither Figure 1 ('Factors Implicated in Patient Involvement in Treatment Decisions') nor Figure 2 ('Continuum of Characteristics Associated with Proactive and Non-Proactive Approaches to Involvement in Treatment Decisions') refer to level of education. We suggest that focusing on education alone would oversimplify the issue as this is not the sole predictor of information access or understanding. For example, knowledge/education in one area does not necessarily translate to another; and the desire to seek-out information is not the sole domain of the highly educated. In the Manuscript (page 21) we noted (for descriptive purposes) that the 5 seemingly proactive information-seeking patients amongst the sample shared certain characteristics (male, mainly younger, educated to degree level, and highly motivated). However, we have not made any general inferences based on these data, or any other data concerning the level of education of any of the study participants.

COMMENT: In the discussion you state that Myeloma participants appear to find this the greatest – is that a reflection on the amount and types of treatment rather than a watch and wait pathway? Your findings may need to reflect the weight that you place on this in the discussion by providing a contrast.

RESPONSE: We have amended the text (Manuscript page 20, lines 13-14) to now read: 'Participants with myeloma, who had most decisions to make, due to the nature of their cancer...'

COMMENT: In your discussion you state that they struggle to weigh up the risks, adverse quality of life and prognostic uncertainty – is this reflected in your findings/quotations?

RESPONSE: This point in the discussion concerning patients with myeloma reflects the findings as reported in Manuscript pages 17-18, reproduced below: 'Disease progression resulted in some patients with myeloma feeling overwhelmed when faced with difficult treatment decisions, and unable to choose between options. Factors compounding this included the intensive nature of proposed treatments (such as stem cell transplant) and their impact on quality of life; the limited "returns" that some treatments seemed to offer, compared to the consequences of associated risks, such as infection; and the uncertainty and unpredictability of outcomes. 'Honestly, my head is exploding with all this...it's just like a big crushing thing to me...I think I am quite strong but this is doing me in' (P35) 'the big one [decision] was the second stem cell transplant...I was really struggling to make the decision as to whether I wanted to go for it' (P18) 'they [doctors] sort of said, well the average remission after the stem cell transplant is, I think

	either 12-24 months, or 18-24 months, something like that, and there was I thinking, right I'm going for the 10-year option, so that was quite a shock' (P18) 'Myeloma is a very individual disease. You get the same treatment, same, same this, same that but you have different outcomes and things.' (P28) 8 COMMENT: Line 32 – the argument flow in this paragraph seems to not be attached to the previous paragraph or the one below. Consider removing as this is clinician TDM not patient TDM. RESPONSE: We are puzzled by this comment, as there is a link to the statement in the previous paragraph, that patients in the study with CLL 'expressed dissatisfaction that they had not been given the opportunity to fully consider, and accept, or decline, treatment' and the following paragraph (which the reviewer refers to), highlighting physicians' perspectives, that where there is good evidence for the selection of a specific treatment, then 'In such instances, the decision is not one of selecting between options, but rather whether the patient chooses to accept or decline treatment.' We suggest that one of the purposes of the discussion is to draw in opinions from other important stakeholders, with reference to the broader research literature and evidence from other studies. Response: I would then suggest that this paragraph is worked to give the emphasis that this is also found in the systematic review as to a reason why this is the case. The paragraph may start with similar this issue has been found within.... To shed light on the complexity of this interactionAs this paragraph gets specific to CLL. COMMENT: Page 20 line 19 the distinction between age, educated level and highly motivated is not explored in the findings. Consider updating the findings to reflect the discussion. RESPONSE: Thank you for your comment. To clarify, our study was not designed to conduct in-depth exploration of inter-relationships between specific factors such as age, educational attainment, and motivation; the small-scale, purposive, qualitative sample employed in our study would not enable us to imply that such relationships exist, or make generalisations based on the findings. We note, however, for descriptive purposes, that the patients whose information seeking activities might be regarded as proactive (see Manuscript Figure 2, as well as Theme 3), who were in their 50s and early 60s, were comparatively younger than the majority of patients included in the study (median age 67 years). COMMENT: Next paragraph – is it that they didn't want to engage, or do they just not understand the complexity – it may not always be about health literacy because it is a very complex science and TDM. Agree with Line 38/39. Line 58 and on progression of disease. RESPONSE: Thank you for your comment. As the reviewer notes, many patients did not want to be involved in TDM, and were happy to trust clinicians to make decisions on their behalf, while others indicated that they might have adopted a more active role in TDM, but perceived the complex nature of their haematological cancer as a deterrent. COMMENT: Clinical Implications: The clinical implications section is weak and should give recommendation that would have clinical utility e.g. underpinned by an assessment overtime of the patient in order to provide the most optimum conditions for the transfer of information. RESPONSE: Thank you for your comment. We agree that assessment over time of the patient is of central importance and thus we had already included in the Manuscript (page 23, lines 23-
--	--

	24) a sentence to that effect: 'Our findings suggest that our interviewees varied in their preference for involvement in TDM according to intrinsic, contextual, and disease-related factors, requiring clinicians to assess individuals' preferences for engagement at multiple time points.' In order to further underscore this recommendation, we have expanded this sentence: 'Our findings suggest that our interviewees varied in their preference for involvement in TDM according to intrinsic, contextual, and disease-related factors, requiring clinicians to assess individuals' preferences for engagement at multiple time points over the course of their haematological cancer pathway.' (Manuscript page 23, lines 23-24). COMMENT: In summary the results section required reviewed. Multiple quotes do give a sense of the variety of participant response but may not always be supported by the wrap around writing and at time can look conflicting in nature. The thematic representation of the results requires review. RESPONSE: Thank you for this final comment which refers to issues raised elsewhere which we hope we have addressed/resolved through our responses and amendments to the Manuscript.
--	---

VERSION 2 – AUTHOR RESPONSE

We are grateful to the reviewer for further commenting on our manuscript. The reviewer has a valid point, and for absolute clarity we have added the following sentence to the end of the Background section of our paper:

"A suite of further papers are under preparation, which address information seeking and sharing, patients' experiences of disease management, and needs for support."